



# PALACE v1.0: Paranal Airglow Line And Continuum Emission model

Stefan Noll[1,2,3], Carsten Schmidt[2], Patrick Hannawald[2], Wolfgang Kausch[4], and Stefan Kimeswenger[4,5]

[1]German Space Operations Center, Deutsches Zentrum für Luft- und Raumfahrt, Oberpfaffenhofen, Germany
[2]Deutsches Fernerkundungsdatenzentrum, Deutsches Zentrum für Luft- und Raumfahrt, Oberpfaffenhofen, Germany
[3]Institut für Physik, Universität Augsburg, Augsburg, Germany
[4]Institut für Astro- und Teilchenphysik, Universität Innsbruck, Innsbruck, Austria
[5]Instituto de Astronomía, Universidad Católica del Norte, Antofagasta, Chile

**Correspondence:** Stefan Noll (st@noll-x.de)

**Abstract.** Below about $2.3\,\mu\mathrm{m}$, the nighttime emission of the Earth's atmosphere is dominated by non-thermal radiation. Excluding aurorae, the emission is caused by chemical reaction chains which are driven by the daytime photolysis and photoionisation of constituents of the middle and upper atmosphere by hard ultraviolet photons from the Sun. As this airglow can even outshine scattered moonlight in the near-infrared regime, the understanding of the Earth's night-sky brightness requires good knowledge of the complex airglow emission spectrum and its variability. However, airglow modelling is very challenging as it would require atomic and molecular parameters, rate coefficients for chemical reactions, and knowledge of the complex dynamics at the emission heights with a level of detail that is difficult to achieve. In part, even the chemical reaction pathways remain unclear. Hence, the comprehensive characterisation of airglow emission requires large data sets of empirical data. For fixed locations, this can be best achieved by archived spectra of large astronomical telescopes with a wide wavelength coverage, high spectral resolving power, and good temporal sampling. Using 10 years of data from the X-shooter echelle spectrograph in the wavelength range from 0.3 to $2.5\,\mu\mathrm{m}$ and additional data from the Ultraviolet and Visual Echelle Spectrograph at the Very Large Telescope at Cerro Paranal in Chile, we have succeeded to build a comprehensive spectroscopic airglow model for this low-latitude site under consideration of theoretical data from the HITRAN database for molecules and from different sources for atoms. The Paranal Airglow Line And Continuum Emission (PALACE) model comprises 9 chemical species, 26,541 emission lines, and 3 unresolved continuum components. Moreover, there are climatologies of relative intensity, solar cycle effect, and residual variability with respect to local time and day of year for 23 variability classes. Spectra can be calculated with a stand-alone code for different conditions, also including optional atmospheric absorption and scattering. In comparison to the observed X-shooter spectra, PALACE shows convincing agreement and is significantly better than the previous, widely used airglow model for Cerro Paranal.

## 1 Introduction

Understanding the radiation spectrum of the Earth's night sky at wavelengths shorter than the thermal emission regime of trace gases such as water vapour requires the knowledge of non-thermal perpetual radiation processes known as airglow or



nightglow. Different atoms and molecules radiate in the Earth's mesopause region between about 75 and 105 km and in the case of atoms also in the upper thermosphere mainly above about 200 km at nighttime due to chemical reactions. The source of
the complex chemistry is usually the energy input of destructive ultraviolet (UV) photons from the Sun, which lead to photolysis and photoionisation in the middle and upper atmosphere at daytime. The resulting radicals and ions (mainly involving oxygen) trigger various reactions and constitute an energy reservoir that is still important at nighttime. Hence, reactions that produce excited states that can be deactivated by photon emission are also present during the night, when scattered sunlight does not disturb.

The different emission processes produce a highly structured spectrum consisting of various emission lines, bands, and unresolved (pseudo-)continua (e.g., Osterbrock et al., 1996; Rousselot et al., 2000; Cosby et al., 2006; Khomich et al., 2008; von Savigny, 2017; Noll et al., 2024a). The ro-vibrational bands of the electronic ground state of the hydroxyl (OH) radical are particularly prominent from the visual to about 2.3 μm with the highest intensities between about 1.5 and 1.8 μm. There are also several important bands of molecular oxygen ($O_2$) in a similar range (0.762, 0.865, 1.27, and 1.58 μm). Weak $O_2$
bands related to high electronic excitation are present in the near-UV and blue range. Various weak bands of iron monoxide (FeO) form a broad emission structure near 600 nm, whereas the near-infrared (near-IR) (pseudo-)continuum with a prominent peak near 1.51 μm appears to be dominated by the hydroperoxyl ($HO_2$) radical. The most crucial atomic airglow lines are located between about 500 and 800 nm including prominent atomic oxygen (O) emission at 558, 630, 636, and 777 nm, the sodium Na D doublet at 589 nm, and the nitrogen (N) doublet at 520 nm. Most emissions originate in the mesopause region in
a relatively narrow hight range. Exceptions are the ionospheric high-altitude lines at 520, 630, 636, and 777 nm.

The variability of airglow is also very complex as there are sources of perturbations with a wide range of time scales and the various emission lines show individual responses depending on the involved chemical species, relevant atomic or molecular parameters, and the vertical emission distribution. The underlying atmospheric dynamics is strongly driven by wave-like variations such as solar tides with preferred periods of 12 and 24 h, gravity waves with periods of minutes to hours, and
planetary waves with periods from days to weeks (e.g., Forbes, 1995; Fritts and Alexander, 2003; Smith, 2012). The interaction of these waves, additional instabilities, and the impact of winds lead to diurnal variability patterns that depend on the season and the observing site. With the effect of airglow chemistry, the resulting airglow climatologies can significantly differ for the investigated emission processes (e.g., Takahashi et al., 1998; Shepherd et al., 2006; Gao et al., 2011; Shepherd, 2016; Reid et al., 2017; Hart, 2019; Noll et al., 2023b, 2024a). Airglow radiation also shows a clear response to the varying solar activity,
which especially affects the influx of hard UV photons. The activity cycle of about 11 years can therefore be well recognised with an amplitude depending on the chemical species and excited state (e.g., Reid et al., 2014; Gao et al., 2016; Hart, 2019; Perminov et al., 2021; Schmidt et al., 2023; Noll et al., 2023b, 2024a).

The modelling of airglow emissions is challenging. Global dynamical models with included airglow chemistry (e.g., Yee et al., 1997; Gelinas et al., 2008; Grygalashvyly et al., 2014; Plane et al., 2015; Noll et al., 2024a) or kinetic models for
specific emission processes (e.g., Dodd et al., 1994; Funke et al., 2012; von Savigny et al., 2012; Panka et al., 2017; Noll et al., 2018; Haider et al., 2022) rely on the knowledge of the chemical composition at all relevant heights, geographic locations, and times, although the implemented dynamics and chemistry can have significant uncertainties. In particular, the impact of





gravity waves with their relatively small spatial scales on the global dynamics is difficult to model and the rate coefficients of many chemical reactions and collisional relaxation processes show significant uncertainties. Sometimes there are only rough guesses as suitable data do not exist. Even measured profiles of important species can be relatively uncertain if the retrieval also depends on an airglow model as in the case of the crucial O concentration (e.g., Mlynczak et al., 2018; Panka et al., 2018; Zhu and Kaufmann, 2018). Theoretical models are important for a better understanding of airglow emission processes. However, there are too many uncertainties to calculate comprehensive airglow spectra with realistic fluxes. Hence, the characterisation of airglow emission has to be mainly empirical and therefore requires large amounts of spectroscopic measurements.

An important application of spectroscopic airglow models is the derivation of the wavelength-dependent sky brightness, where airglow is a major component (e.g., Leinert et al., 1998). In particular, this is relevant for the efficient scheduling of observations at large astronomical facilities, the design of astronomical instruments, and data processing removing atmospheric signatures. Such a model was developed for the Very Large Telescope (VLT) of the European Southern Observatory (ESO) at Cerro Paranal (24.6° S, 70.4° W) in Chile (Noll et al., 2012; Jones et al., 2013; Noll et al., 2014). Until now, this "Sky Model" is still the most popular model of this kind. Masana et al. (2021) released an alternative sky brightness model but without internal airglow calculations for different conditions.

The airglow component of the ESO Sky Model consists of a list of 4,764 emission lines in the wavelength range from 0.3 to 2.5 µm. Up to 0.925 µm, the list is based on the line identifications of Cosby et al. (2006) in the catalogue of Hanuschik (2003) derived from line measurements in composite spectra of the high-resolution Ultraviolet and Visual Echelle Spectrograph (UVES; Dekker et al., 2000) at the VLT. At longer wavelengths, the simple OH level population model of Rousselot et al. (2000) described by only two (pseudo-)temperatures (190 and 9,000 K) was used and scaled to the Cosby et al. (2006) intensities in the overlapping wavelength region. The population model was combined with Einstein-$A$ coefficients for photon emission from the HITRAN2008 database (Rothman et al., 2009). The latter was also used for obtaining line lists for the $O_2$ bands at 1.27 and 1.58 µm assuming a temperature of 200 K. The intensities of the $O_2$ bands were scaled to be consistent with those of neighbouring OH bands using a small sample of near-IR spectra of the VLT X-shooter echelle spectrograph (Vernet et al., 2011) and considering atmospheric absorption. The airglow emissions were classified using the green O line at 558 nm, the red O lines at 630 and 636 nm, the Na D doublet at 589 nm, the OH bands in the range from 642 to 858 nm, and the $O_2$ band at 865 nm as references. The intensities of all lines belonging to each of the five classes were multiplied by correction factors to achieve that the intensities of the corresponding reference features match values that were derived from measurements in a sample of 1,186 low-resolution long-slit spectra taken with the VLT FOcal Reducer and low dispersion Spectrograph 1 (FORS 1; Appenzeller et al., 1998) and processed by Patat (2008). The spectra were taken between April 1999 and February 2005 and cover parts of the maximum wavelength range from 365 to 890 nm. The resulting standard values for the reference features correspond to the mean intensities for a solar radio flux at 10.7 cm (Tapping, 2013) of 129 solar flux units (sfu) or 1.29 MJy, where the unit jansky (Jy) equals $10^{-26}$ W m$^{-2}$ Hz$^{-1}$. Based on a linear regression analysis for each class, the model intensities can be adapted to arbitrary solar radio fluxes (although only 95 to 228 sfu were covered by the data). Moreover, variability is considered by a climatological grid that comprises six double months (starting with December/January) and three nightime bins (dividing the night in ranges of equal length). Apart from these 18 scaling factors, the model also provides



the residual variability for each bin. The ESO Sky Model also considers the increase of intensity with increasing zenith angle
due to the change in the projected emission layer width along the line of sight (van Rhijn, 1921).

Apart from five line emission classes, there is also one class related to the unresolved residual continuum after the subtraction
of other radiation components such as zodiacal light or scattered light from the Moon and stars. The reference spectrum for
this airglow-related component was derived from the sample of FORS 1 spectra and the small number of X-shooter spectra
in windows without strong line emission. The uncertainties are relatively high. For the variability model, only the variation at
543 nm in the FORS 1 data was considered. By means of other components of the Sky Model, the airglow line and continuum
emission is also corrected for absorption and scattering (mainly in the lower atmosphere) depending on the zenith angle and
the season (or the amount of water vapour).

    Despite the described complexity, the airglow component of the ESO Sky Model shows clear limitations. The variability
model is only based on about $10^3$ spectra with varying wavelength coverage in the range from 365 to 890 nm. The line list is
an unsatisfactory mixture of measurements and simple models from different sources. The continuum determination suffered
from the low resolution of the FORS 1 spectra and the calibration uncertainties related to the few early X-shooter spectra that
were used. Hence, a significant improvement is possible. In the meantime, the airglow emissions above Cerro Paranal were
studied in more detail based on consistent spectroscopic data sets. UVES spectra covering 15 years (starting in April 2000)
in the wavelength range from 0.57 to 1.04 μm were used to investigate long-term OH variations (Noll et al., 2017), the OH
ro-vibrational level populations (Noll et al., 2018, 2020), and the variability of the potassium (K) emission at 770 nm (Noll
et al., 2019). The amount and quality of X-shooter spectra covering the full wavelength range from 0.3 to 2.5 μm increased in
the course of the years. Starting with a few hundred spectra to study OH and $O_2$ emissions (Noll et al., 2015, 2016), then using
a few thousand spectra to investigate the FeO-related pseudo-continuum (Unterguggenberger et al., 2017), finally resulted in
studies of OH climatologies (Noll et al., 2023b) and the airglow continuum (Noll et al., 2024a) based on the order of $10^5$
spectra covering 10 years from October 2009 to September 2019.

We used the large X-shooter data set to measure the variations of all remaining significant airglow emissions in order to
characterise the full airglow spectrum. Combined with theoretical data such as level energies, Einstein-$A$ coefficients, and re-
combination coefficients as well as fits of level populations, we could create a list of 26,541 lines with attributed climatologies.
Based on the work of Noll et al. (2024a), we also added three continuum components with the corresponding variations. In
total, reference climatologies that cover nocturnal changes, seasonal variations, the response to solar activity, and residual vari-
ations were derived for 23 variability classes. This Paranal Airglow Line And Continuum Emission (PALACE) model (Noll
et al., 2024b), which can be used to calculate airglow spectra for different conditions by means of an accompanying Python
code, will be discussed in the following. The basic structure and algorithm is explained in Sect. 2. Section 3 provides details
on the data set for the analysis. Section 4 discusses the creation of the line lists with reference intensities for the different
chemical species. The pseudo-continuum template spectra are the topic of Sect. 5. The reference climatologies are discussed
in Sect. 6. The performance of PALACE is analysed in Sect. 7, also in comparison with the ESO Sky Model. Finally, we draw
our conclusions in Sect. 8.



**Table 1.** PALACE model parameters and their default values

| Name | Description | Unit | Type | Constraints | Default |
|------|-------------|------|------|-------------|---------|
| species | specific molecule/atom or all | – | str | all, OH, O2, HO2, FeO, Na, K, O, N, H | all |
| z | zenith angle | deg | float | $\geq 0.$ and $< 90.$ | 0. |
| mbin | month number[a] or 0 for all | – | int | [0, 12] | 0 |
| tbin | local time[b] bin[c] or 0 for all | – | int | [0, 12] | 0 |
| | local time[b] | h | float | [-6., 6.] or ([0., 6.] and [18., 24.]) | |
| srf | solar radio flux at 10.7 cm | sfu[d] | float | $\geq 0.$ | 100. |
| isair | wavelength in standard air? | – | bool | True (air) or False (vacuum) | True |
| isatm | absorption and scattering? | – | bool | True (with effects) or False | True |
| pwv | precipitable water vapour[e] | mm | float | $\geq 0.$ | 2.5 |
| lammin | minimum wavelength | μm | float | $\geq 0.3$ | 0.3 |
| lammax | maximum wavelength | μm | float | $\leq 2.5$ | 2.5 |
| dlam | constant wavelength step | μm | float | $\geq 10^{-7}$ (1e-7) | 1e-4 |
| resol | constant resolving power[f] | – | float | $> 0.$ | 1e3 |
| outdir | output directory[g] | – | str | write access required | test/ |
| outname | output file name[h] | – | str | write access required | palace_test |
| specsuffix | suffix of spectrum file | – | str | fits for FITS, free suffix for ASCII format | fits |
| showplot | plot of spectrum on screen? | – | bool | True (opens plot window) or False | False |

[a] 1 for January and 12 for December.

[b] Solar mean time at Cerro Paranal, i.e. 70.4° W.

[c] 1 for 18–19 h and 12 for 5–6 h.

[d] 1 sfu = $10^4$ Jy.

[e] Only relevant if isatm is True.

[f] Fixed ratio of FWHM of Gaussian line spread function and wavelength.

[g] Will be created if required.

[h] Supplemented by suffixes .par for a parameter file, .fits for a FITS table, or anything else for an ASCII table.

## 2  Model overview

The core of PALACE are three tables in Flexible Image Transport System (FITS) format which provide the reference line
intensities, the continuum component spectra, and the climatologies describing the variability. The three tables are linked via
the defined airglow variability classes. For certain applications of the model, these data might already be sufficient. However,



the calculation of airglow spectra for different conditions is not trivial. Therefore, we wrote a dedicated code that consists of the Python module 'palace' and an optional script for execution ('palace_run.py'). For better performance, we also included optional Cython types, which requires the compilation of the module if desired. For more details on the installation and execution of PALACE, we refer to the README file (Noll et al., 2024b).

The model output depends on 16 parameters which are listed in Table 1. Default values of these parameters are provided by a standard configuration file. They are also listed in the table. The model parameters can be modified by providing a different parameter file and/or changing individual values via command line parameters in the case of the script or manipulating a Python dictionary in the case of the use of a Python shell or similar environments. The model writes output files in the `outdir` directory, which is created recursively if it does not exist. Relative paths are relative to the working directory. Usually, two

files are produced that are named as `outname` plus different suffixes. First, a list of the selected parameter values is written. This file is marked by '.par'. Second, the airglow spectrum is written. If `specsuffix` is set to 'fits' (default), a FITS table with a '.fits' suffix is created. In all other cases, an ASCII file with the given suffix is produced. In any case, the output spectrum has the three columns 'lam' for the wavelength in micrometres, 'flux' for the photon flux in rayleighs per nanometre ($\mathrm{R\,nm^{-1}}$), and 'dflux' for the residual variability not explained by the model with the same unit. In comparison, the ESO Sky

Model uses $\mathrm{photons\,s^{-1}m^{-2}\mu m^{-1}arcsec^{-2}}$ as radiance unit (Noll et al., 2012). This unit can be obtained from $\mathrm{R\,nm^{-1}}$ by the multiplication of 18.704 as 1 R equals $10^{10}/(4\pi)\,\mathrm{photons\ s^{-1}m^{-2}sr^{-1}}$. The wavelength grid of the output spectrum is defined by the parameters `lammin`, `lammax`, and `dlam`. All values have to be in micrometres. Wavelengths lower than $0.3\,\mu\mathrm{m}$ or larger than $2.5\,\mu\mathrm{m}$ would be outside the valid model range. Moreover, wavelength steps shorter than $0.1\,\mathrm{pm}$ are not accepted. Such values would be distinctly lower than the typical natural width of airglow lines of a few picometres. Apart from writing

the spectrum to a file, there is also the option to show it in a Python plot window. In order to enable this option, `showplot` needs to be set to 'True'.

In order to adapt the reference line intensities and continuum fluxes to specific conditions, PALACE calculates scaling factors that depend on 23 different variability classes (see Sect. 6), month, local time, solar activity, and zenith angle. The month is provided via the input parameter `mbin`, where '1' refers to January and '12' to December. It is also possible to request an

annual mean by setting `mbin` to '0'. In a similar way, the local time (LT) parameter `tbin` can be set to values of '0' for the entire night (for a certain month or the entire year depending on `mbin`) or '1' for 18:00 to 19:00 LT to '12' for 05:00 to 06:00 LT. The times refer to the solar mean time at Cerro Paranal, i.e. the Universal Time is corrected by a fixed amount determined by the longitude of $70.4°$ W. If floating point numbers are provided as `tbin`, they are directly interpreted as LTs in hours and the corresponding LT bin is chosen. In this case, the valid maximum range is either 18.0 to 24.0 and 0.0 to 6.0 or

$-6.0$ to 6.0. In fact, the night length constrained by a minimum solar zenith angle of $100°$ is usually shorter, especially during austral summer. For month centres, the latest start and the earliest end of the night are in January ($-4.36$) and December ($4.31$), respectively. If a specific LT or a whole bin does not have nighttime contribution, a warning message is returned. In this case, the model still works but only uses unreliable extrapolated values.

The parameters `mbin` and `tbin` determine the climatological correction factor $f_0$ in comparison to the annual nocturnal

mean intensity for each variability class. However, these correction factors are only valid for a fixed solar radio flux of $100\,\mathrm{sfu}$.



For other levels of solar activity given by `srf`, the factors are adapted using the results of a linear regression analysis depending on `mbin` and `tbin`. The final climatological correction factors $f$ are then calculated by

$$f(\texttt{mbin}, \texttt{tbin}, \texttt{srf}) = f_0(\texttt{mbin}, \texttt{tbin})\,(1 + 0.01\,m_{\mathrm{SCE}}(\texttt{mbin}, \texttt{tbin})\,(\texttt{srf} - 100)), \tag{1}$$

where $m_{\mathrm{SCE}}$ is the regression slope relative to $100\,\mathrm{sfu}$. For the `srf` parameter, daily and monthly solar radio fluxes at $10.7\,\mathrm{cm}$

can be obtained from `https://spaceweather.gc.ca/`. Note that the analysis for PALACE was based on centred 27-day averages (see Noll et al., 2017, for a discussion). Further details on the solar cycle effect and its analysis are provided in Sect. 6. The climatological correction factors $\sigma_f$ for the standard deviation of the residual variability that is given in the 'dflux' column in the output spectrum are calculated from the reference values $\sigma_{f,0}$ relative to the bin-specific $f_0$ by

$$\sigma_f(\texttt{mbin}, \texttt{tbin}) = f_0(\texttt{mbin}, \texttt{tbin})\,\sigma_{f,0}(\texttt{mbin}, \texttt{tbin}), \tag{2}$$

i.e. there is no explicit solar activity term. The reason is the calculation of the residual fluxes in the analysed data set by the subtraction of the climatological model represented by Eq. (1), which already contains this effect.

By default, all intensities and fluxes are given for zenith. This can be changed by means of the input parameter `z`, which provides the zenith angle of the line of sight in degrees. An increase of `z` also increases the projected layer width and, hence, the intensity or flux. For a thin layer, the reciprocal correction factor can be calculated by

$$f_{\mathrm{vR}}^{-1} = \sqrt{1 - \left(\frac{R\sin(\texttt{z})}{R + h}\right)^2} \tag{3}$$

(van Rhijn, 1921), where $R$ is the Earth's radius and $h$ corresponds to the height of the layer above the ground. For $R$, we use the mean radius of $6371\,\mathrm{km}$. The effective layer heights depend on the emission process. Rough reference values are given in Table 2. The heights are not constant and can easily vary by several per cent. However, this is not crucial as the resulting $f_{\mathrm{vR}}$ are very similar for a wide height range as long as the line of sight is not too close to the horizon. Results for zenith angles

of $60°$ or $70°$ are still quite robust. Table 2 shows a wide range of reference layer heights from $81\,\mathrm{km}$ for $HO_2$ to $300\,\mathrm{km}$ for most O lines. More information on these values will be given in Sects. 4 and 5. Note that the van Rhijn effect is not considered in the case of atomic hydrogen (H) as the emissions are caused by fluorescence, which does not produce a well-defined layer (see Sect. 4.7).

At the high altitudes of the airglow layers, the atmospheric density is very low, which results in optically thin environments

for most emission lines. However, for ground-based observations at Cerro Paranal, the airglow photons have to pass much denser atmospheric layers, which can lead to significant absorption and scattering. PALACE considers these effects if `isatm` is 'True', which is the default setting. The level of molecular absorption is very different in the model wavelength range. The transmission $T$ ranges from nearly 1 close to $400\,\mathrm{nm}$ to almost 0, especially in parts of the strong near-IR water vapour absorption bands at about 1.4 and $1.9\,\mu\mathrm{m}$ (e.g., Smette et al., 2015). For the consideration of atmospheric absorption, PALACE

applies a similar approach as was also used for the data analysis, but in the reverse direction. The method is essentially described by Noll et al. (2015). The PALACE data for the emission lines and continuum components include reference transmission values $T_{\mathrm{ref}}$ for zenith and typical conditions at Cerro Paranal. In fact, the latter equal the standard conditions of the ESO Sky





**Table 2.** Summary on chemical species in PALACE

| Species | Mol mass[a] kg mol$^{-1}$ | Lines[b] | Continua[c] | Layer height[d] km | Temperature[a] K |
|---|---|---|---|---|---|
| OH | 0.017 | 22,058 | 0 | 87 | 190 |
|  | 0.019 | 47 | 0 |  |  |
| O$_2$ | 0.032 | 3,398 | 1 | 89 (a-X), 94 (other) | 190 |
|  | 0.033 | 13 | 0 |  |  |
|  | 0.034 | 678 | 0 |  |  |
| HO$_2$ | 0.033 | 0 | 1 | 81 | 190 |
| FeO[e] | 0.072 | 0 | 1 | 88 | 190 |
| Na | 0.023 | 2 | 0 | 92 | 190 |
| K | 0.039 | 2 | 0 | 89 | 190 |
| O | 0.016 | 313 | 0 | 97 (558 nm), 250 (630/636 nm), 300 (other) | 190 (558 nm), 1,000 (other) |
| N | 0.014 | 2 | 0 | 250 | 1,000 |
| H | 0.001 | 28 | 0 | −1 | 1,000 |
| all |  | 26,541 | 3 |  |  |

[a]  For calculation of line width by Doppler broadening, not relevant for continuum components.

[b]  Number of resolved lines in model, 0 for unresolved continua.

[c]  Number of continuum components.

[d]  For calculation of emission increase with increasing zenith angle (van Rhijn effect), not applicable to H (-1).

[e]  Additional molecules probably contribute to the corresponding continuum component.

Model (Noll et al., 2012) with a fixed amount of precipitable water vapour (pwv) of 2.5 mm, which is close to the long-term median (Holzlöhner et al., 2021). The airglow-specific $T_{\mathrm{ref}}$ values were derived from a transmission spectrum calculated with
the Line-By-Line Radiative Transfer Model (LBLRTM; Clough et al., 2005) with maximum resolving power of $4 \times 10^6$ in order to resolve individual airglow lines. PALACE adapts the reference values depending on the input parameters z and pwv in Table 1. The amount of water vapour can be varied independently as it is the most important absorber in the model range and is also highly variable. According to Noll et al. (2015), the $T$ values are approximated from $T_{\mathrm{ref}}$ by

$$T = T_{\mathrm{ref}}^{\left(1 + (r_{\mathrm{pwv}} - 1)\, f_{\mathrm{H_2O}}\right) X}, \tag{4}$$



where $r_{\mathrm{pwv}}$ is the ratio of the selected `pwv` and the reference of $2.5\,\mathrm{mm}$, $f_{\mathrm{H_2O}}$ corresponds to the fraction of water vapour with respect to the total optical depth, and the airmass $X$ is calculated by

$$X = \left( \cos(\mathrm{z}) + 0.025\,\mathrm{e}^{-11\cos(\mathrm{z})} \right)^{-1} \tag{5}$$

(Rozenberg, 1966). The values of $f_{\mathrm{H_2O}}$ are also provided by the PALACE data files. They were derived by means of the comparison of transmission spectra for 1 and $5\,\mathrm{mm}$, which enclose a major fraction of the `pwv` values for clear sky conditions

at Cerro Paranal (Holzlöhner et al., 2021). Note that the ESO Sky Model uses a large library of transmission data instead of the discussed approximation. This is more accurate but only for `z` and `pwv` with data in the library.

Scattering of photons at molecules (Rayleigh scattering) and aerosol particles (Mie theory) also changes the airglow brightness. As airglow is present at the entire sky, photons can be scattered out of and into the line of sight, which decreases the effective exinction compared to the case of point sources. Noll et al. (2012) modelled this effect for Cerro Paranal and derived

recipes that depend on `z`. We also use these results combined with the same Rayleigh and aerosol extinction curves, which describe the wavelength dependence of the extinction for point sources. For aerosol scattering, the basis is the standard curve for Cerro Paranal from Patat et al. (2011). The change of airglow emission by scattering is usually small (especially in the near-IR). However, at near-UV and blue wavelengths combined with high zenith angles, impacts larger than 10% are possible. Close to the zenith, scattering slightly increases the airglow brightness for clear sky conditions as the intensity minimum at

zenith is partly filled by photons from brighter regions at higher `z` (van Rhijn effect).

The PALACE data files contain wavelengths in vacuum. However, spectroscopic data are often given for standard air. Hence, PALACE supports both options. If the input parameter `isair` is set to the default 'True', air wavelengths $\lambda_{\mathrm{air}}$ are provided by calculating

$$\lambda_{\mathrm{air}} = \frac{\lambda_{\mathrm{vac}}}{n}, \tag{6}$$

where $n$ is the refractive index. It is approximated by

$$n = 1 + 10^{-8} \left( 8342.13 + \frac{2406030}{130 - \lambda^{-2}} + \frac{15997}{38.9 - \lambda^{-2}} \right) \tag{7}$$

with $\lambda$ in micrometres (Edlén, 1966). This approach is consistent with Noll et al. (2012).

After the application of the scaling factors from the climatologies, van Rhijn effect, and atmospheric extinction (absorption and scattering) to the reference data, line intensities need to be converted to line fluxes in order to obtain spectra. For this

purpose, we assume that the natural line shape is similar to a Gaussian function. This is a good approximation as thermal Doppler broadening predominates due to the very low air densites at the emission heights (see also Noll et al., 2012). In this way, the line-specific width of the Gaussian $\sigma_\lambda$, which is of the order of a picometre (except for H), can be derived by

$$\sigma_\lambda = \lambda_0 \sqrt{\frac{k_{\mathrm{B}} N_{\mathrm{A}} T_{\mathrm{kin}}}{M\,c^2}}, \tag{8}$$

where $k_{\mathrm{B}}$, $N_{\mathrm{A}}$, and $c$ are the Boltzmann constant, Avogadro constant, and the speed of light, respectively. Moreover, the

line-dependent central wavelength $\lambda_0$, the kinetic temperature $T_{\mathrm{kin}}$, and the molecular weight $M$ need to be provided. These





parameters are listed in the line table of the model. Molecular weight and temperature are also summarised in Table 2. In the case of $T_{\text{kin}}$ only two values are used. For the mesopause region, we take 190 K, which agrees well with the mean temperature profile at Cerro Paranal (Noll et al., 2016) measured with the satellite-based Sounding of the Atmosphere using Broadband Emission Radiometry (SABER) instrument (Russell et al., 1999). For the upper thermosphere, 1,000 K is a reasonable assump-

tion for the neutral and ion temperatures (e.g., Bilitza et al., 2022). The temperatures can significantly vary in reality. However, this is not crucial due to the square root in Eq. (8) and the visibility of natural line widths only in spectra with extremely high resolution.

The Gaussian for each scaled line is calculated for the desired wavelength grid given by `lammin`, `lammax`, and `dlam` and under consideration of `isair`. Then, all Gaussians are summed up. Furthermore, the scaled continuum components are added

and the sum spectrum is mapped to the selected wavelength grid. The combination of the resulting line and continuum spectra constitutes the total emission. The spectrum for the residual variability is calculated in the same way. In principle, the simple summation would require that overlapping emissions show variations which are fully correlated. In reality, this can be different (see Sect. 6). Emissions might even be partly anti-correlated, which would lead to a decreased standard deviation compared to the individual variations. Hence, the output variability represents a maximum. Nevertheless, this is only a minor issue as

adjacent lines are often of similar type and most wavelength ranges are dominated by a single variability class. Moreover, it would be very challenging to derive effective correlations as they can depend on the specific perturbation.

The final step of the spectrum calculation in PALACE considers the spectral resolving power $\lambda/\Delta\lambda$. It is a constant for the entire wavelength range set by the parameter `resol`. In principle, `resol` is independent of the wavelength grid with the step size `dlam` but real spectra usually have several pixels per resolution element. For the default values of `resol` and `dlam`

in Table 1 of 1,000 and 0.1 nm, the number of pixels is 3 at 0.3 µm and 25 at 2.5 µm. The limited resolution is simulated by the convolution of the combined line and continuum spectrum with wavelength-dependent Gaussians where $\Delta\lambda$ corresponds to the full width at half maximum (FWHM), which corresponds to $\sqrt{8\ln 2}\,\sigma_\lambda$. The half size of the kernel is fixed to $5\,\sigma_\lambda$. As the convolution near `lammin` and `lammax` requires additional airglow data, the previous steps of the calculation of the output spectrum are performed with an extended wavelength grid. Note that the use of only Gaussians for the simulation of the

line-spread function is simpler than in the case of the ESO Sky Model, which also offers boxcar and Lorentzian components. The reproduction of complicated instrumental line-spread functions is not in the focus of PALACE.

The final spectrum for the default parameters as listed in Table 1 is plotted for the wavelength range from 0.32 to 2.34 µm in Fig. 1. The most relevant emission features have already been discussed in Sect. 1. In order to better distinguish the contributions from the different chemical species, the figure also shows the specific spectra of OH, $O_2$, atomic lines, and unresolved

continua (including $HO_2$ and FeO-like emissions). Spectra for each chemical species can also be calculated by PALACE. This is possible by setting the parameter `species` to the corresponding empirical formula. By default, `species` is set to 'all'. Figure 1 also illustrates the impact of `isatm`. For 'False', the dotted curve shows the spectrum without atmospheric absorption and scattering, which indicates a particular high emission increase for the ranges of the already mentioned water vapour bands near 1.4 and 1.9 µm as well as the $O_2$ bands at 0.762 and 1.27 µm. Without atmospheric extinction, the total emission of the

reference spectrum amounts to 897 kR, otherwise 603 kR.



**Figure 1.** Reference PALACE spectrum for default model parameters provided in Table 1 in the wavelength range from 0.32 to 2.34 μm divided into five sections with different radiance ranges. The combined spectrum for all components (black solid lines) is also provided without atmospheric absorption and scattering (black dotted lines). Moreover, the contributions of OH (blue), $O_2$ (cyan), atoms (red), and the pseudo-continuum related to $HO_2$, FeO, and other molecules (yellow) are shown.





## 3   Data set

The reference data set for the build-up of the semi-empirical PALACE model consists of 10 years of X-shooter (Vernet et al., 2011) echelle spectra taken for astronomical projects at Cerro Paranal between October 2009 and September 2019. The properties and airglow-opimised processing of these data from the ESO Science Archive Facility were already described in detail by
Noll et al. (2022, 2023b, 2024a). Therefore, we provide here only a brief overview. X-shooter consists of three so-called arms with optimum wavelength ranges (after the evaluation of the overlapping regions) of 0.30 to 0.56 μm (UVB), 0.55 to 1.02 μm (VIS), and 1.02 to 2.48 μm (NIR). These arms are usually operated in parallel, although the exposure times of each arm can be set independently. In any case, X-shooter allows the comparison of the variability of a high number of airglow emission lines of different origin. For the standard width of the entrance slits of each arm, the resulting resolving power is 5,400 (UVB),
8,900 (VIS), and 5,600 (NIR), respectively. This is sufficiently high to separate many strong and weak airglow emissions. On the other hand, line multiplets such as OH $\Lambda$ doublets are rarely resolved. However, this is not an issue as the variations of the components are expected to be similar. The slit widths can be set to different values, which causes a varying spectral resolving power in the data set (e.g. from 3,500 to 12,000 in the NIR arm). However, a much wider range of possibilities is related to the exposure time, which can vary by several orders of magnitude. Hence, the data selection depends on the signal-
to-noise ratio requirements for the airglow emission features in focus. This is a drawback of archived astronomical spectra that were taken for very different targets and purposes. In the same way, residual contaminations of the astronomical targets in the extracted airglow spectra can be very different in strength and wavelength dependence. Therefore, each airglow feature was investigated with an optimum set of spectra. This is not an issue as long as the resulting samples are still large enough to derive climatologies with sufficient quality. We only considered those emissions where this was possible. The maximum useful data
set consists of about 56,000 UVB, 64,000 VIS, and 91,000 NIR spectra. The different numbers are mainly due to a different splitting of exposures in part of the observations. The thorough flux calibration of the resulting one-dimensional spectra based on time-dependent sets of master response curves resulted in a scatter (standard deviation) of only a few per cent in most of the wavelengh range if flux-calibrated standard star spectra are compared. There is also a possible wavelength-dependent systematic offset of up to several per cent due to uncertainties in the theoretical star spectra that were used for the derivation of
the response curves.

The spectral resolving power of X-shooter allows the measurement of many airglow emission lines. However, this is often not possible for faint lines in crowded wavelength regions. Hence, we also used published airglow data from the UVES echelle spectrograph (Dekker et al., 2000) at the VLT. UVES covers a narrower wavelength range than X-shooter of 0.31 to 1.04 μm in maximum if set-ups with different central wavelengths are combined. However, the resolving power is much higher. Depending
on the width of the entrance slit, values between about 20,000 and 110,000 are possible. Hanuschik (2003) measured 2,810 emission features in composite spectra of different set-ups covering the maximum wavelength range. The spectra were created from a small number of long-term exposures taken between June and August 2001. As only observations with standard slits were included, the resulting resolving power is between 43,000 and 45,000. Cosby et al. (2006) identified the airglow lines related to the measured features. For Noll et al. (2012), this catalogue was an important data source for the ESO Sky Model.





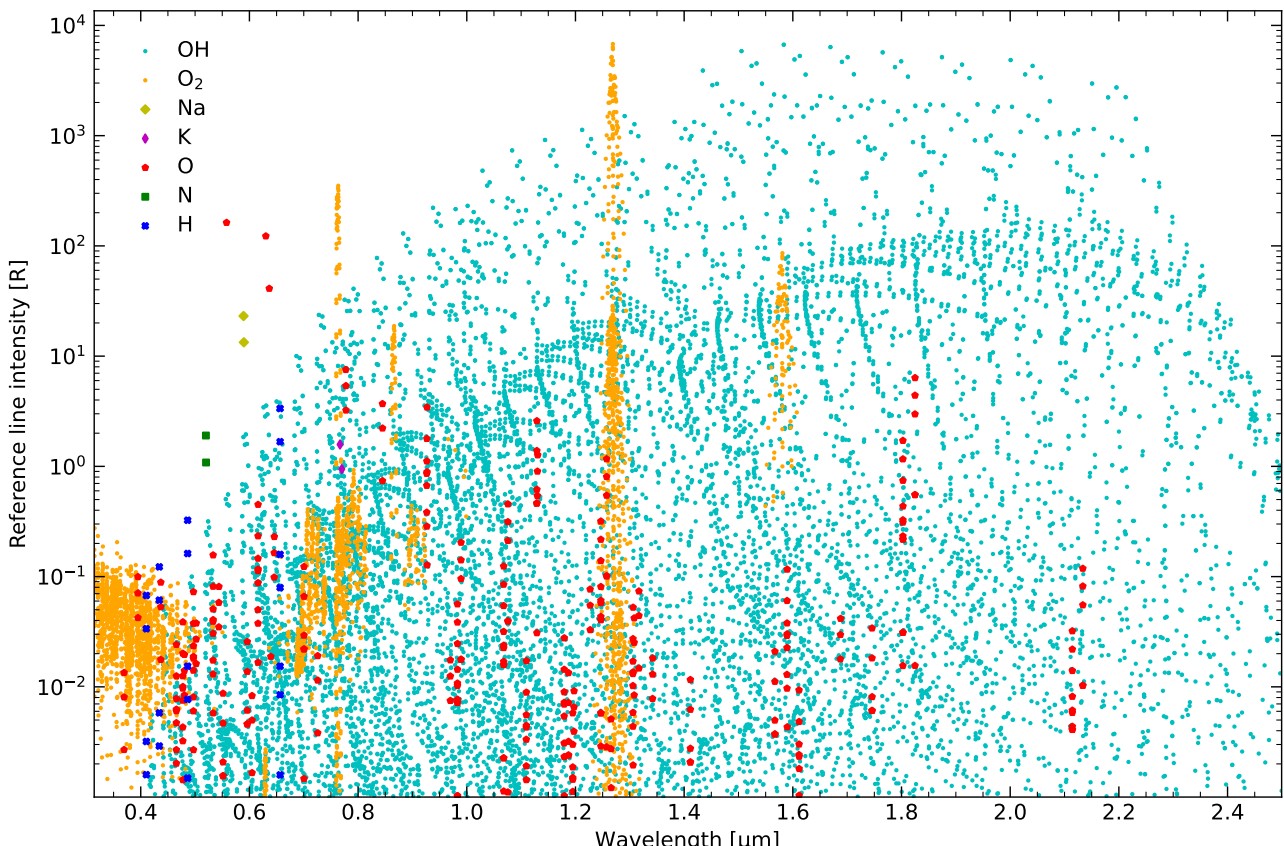

**Figure 2.** Reference line intensities of PALACE model, i.e. annual nocturnal mean values for a solar radio flux of 100 sfu, as a function of wavelength. Atmospheric absorption and scattering is not applied. The 16,047 lines of OH, $O_2$, Na, K, O, N, and H with a minimum intensity of 1 mR are marked by different symbols and colours (see legend).

For PALACE, we focused on faint OH and $O_2$ lines without X-shooter measurements. Another UVES-related data set was prepared by Noll et al. (2017). It consists of about 10,400 spectra taken between April 2000 and March 2015 with the two reddest set-ups with central wavelengths of 760 and 860 nm covering a maximum range from 0.57 to 1.04 µm. A subsample of 2,299 spectra was then used by Noll et al. (2019) to study the intensity and variations of the faint K emission line at 770 nm. Moreover, 533 long-term exposures were selected by Noll et al. (2020) to calculate a high-quality composite spectrum, which was used to study OH populations. The line measurements of both publications were also considered for PALACE.

## 4 Line emission

The list of reference line intensities is an important ingredient of PALACE. It is illustrated by Fig. 2, which shows intensity vs. wavelength for all lines with a strength of at least 1 mR. The included chemical species are marked by different symbols and



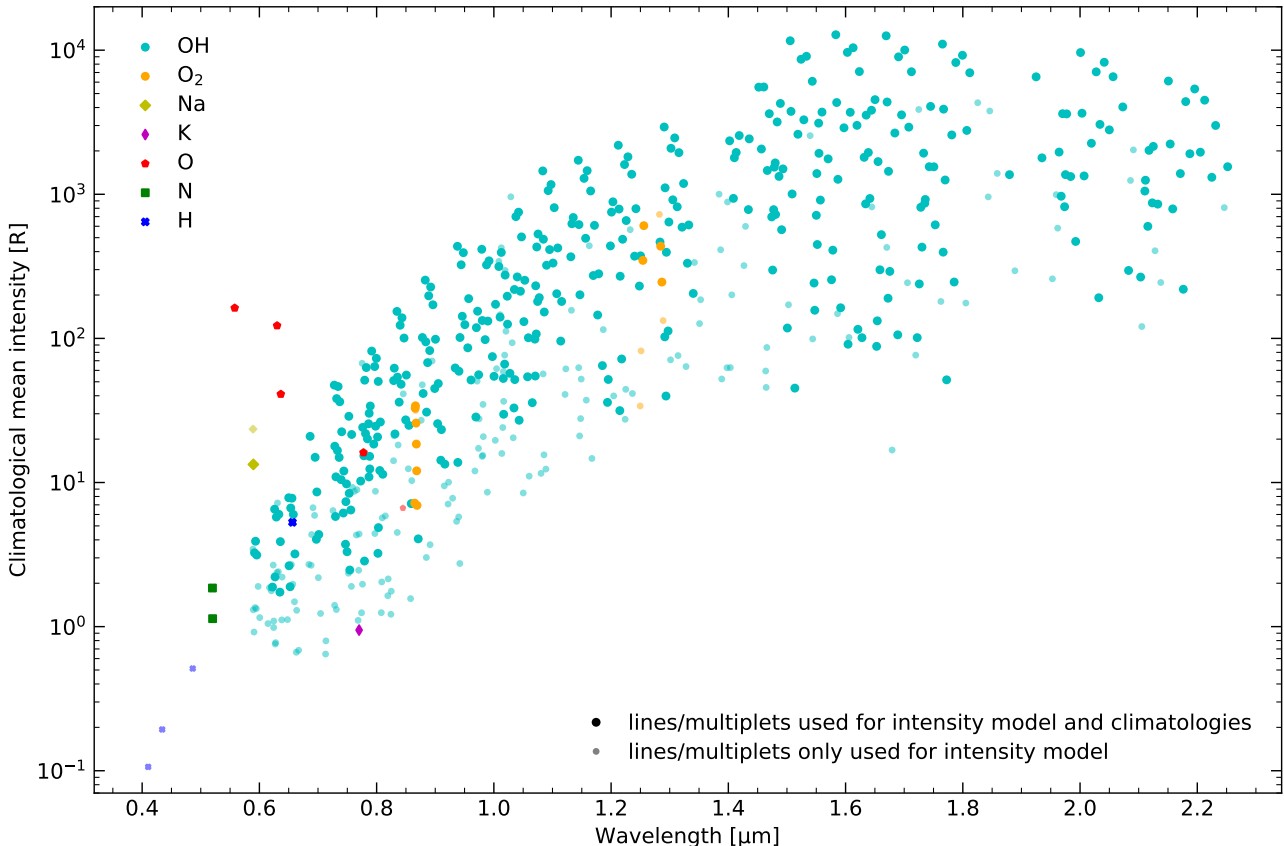

**Figure 3.** Individual lines or multiplets with measured climatologies that were used for the development of the mean intensity model as shown in Fig. 2 (574 cases) and the derivation of the reference climatologies as listed in Table 4 (392 cases, marked by big symbols, see legend). Except for the UVES-based $K\,D_1$ data from Noll et al. (2019), all data originate from X-shooter measurements. Each line or multiplet is identified by the chemical species (different symbols and colours, see legend), annual nocturnal mean intensity for 100 sfu, and wavelength.

colours. The total number of lines for each chemical species (or isotopologue if relevant) is provided in Table 2. Figure and

table clearly show the predominance of OH lines (22,105 of 26,541 lines), followed by $O_2$ and O emissions. In contrast, the atoms Na, K, and N are only present by a doublet. The wavelength distribution of the line intensites can be compared to the spectrum without atmospheric extinction in Fig. 1. A small subset of strong lines is decisive for the visible structures in most wavelength ranges. An exception is the near-UV and blue wavelength regime, where only weak lines are present.

The derivation of reference intensities for 26,541 lines was only possible by the use of theoretical data such as Einstein-$A$

coefficients, level energies, and recombination coefficients. Hence, PALACE is a semi-empirical model. As demonstrated by Fig. 3, the data set with derived climatologies is distinctly smaller. It only consists of 574 intensity values, which are mostly related to OH (544 data points). Except for the UVES-based data for the K line at 770 nm (Noll et al., 2019), all measurements are related to X-shooter. Line intensities from composite spectra without variability information (Cosby et al., 2006; Noll et al.,



2020) are not displayed. Even if a climatology was measured and used to derive a reference intensity, it was not always good

enough to be taken for the variability model (Sect. 6). Hence, the latter was derived from only 392 climatologies (see big symbols in Fig. 3). Note that the intensities were often measured for unresolved multiplets. In the case of OH lines, the plotted intensities are related to $\Lambda$ doublets. Line blends are also relevant for the H lines, the O 777 and 845 nm emissions, and the $O_2$ band at 865 nm. Therefore, these blends also contribute to the difference of data points in Figs. 2 and 3.

Except for the OH lines in the NIR arm (Sect. 3) discussed by Noll et al. (2023b), the X-shooter intensity time series

used for PALACE are new measurements. For lines without existing data set, we used the same approach as described by Noll et al. (2022, 2023b, 2024a). First, the line emission was separated from the underlying continuum, which consists of unresolved airglow emission (see Sect. 5) as well as other contributions to the night-sky brightness such as zodiacal light or scattered moonlight. The separation was achieved by the application of percentile filters with varying percentile (50 to 20 % for increasing density of strong lines) and window width relative to the wavelength (0.008 to 0.04). After the subtraction of

the filtered continuum, fluxes were integrated in wavelength intervals around the target line positions where the width relative to the wavelength depended on the spectral resolution (defined by the X-shooter arm and the width of the entrance slit) and wavelength differences of line components in the case of measured multiplets. For molecules, the line positions originate from the 2020 version of the HITRAN database (Gordon et al., 2022) with revised OH-related calculations based on the measurements of Noll et al. (2020). For atoms, the current version of the National Institute of Standards and Technology

(NIST) database (Kramida et al., 2023) was used.

The resulting line intensities were corrected for the van Rhijn effect as well as atmospheric absorption and scattering. This worked in the same way as discussed in Sect. 2, with the only difference that the reverse factors were applied. While the zenith angle in the middle of an exposure is well determined, the effective `pwv` value of each spectrum needs a measurement. As described by Noll et al. (2022), the `pwv` value was estimated from OH line pairs with very different absorption. The

calibration of the relations was achieved by regular water vapour measurements with a microwave radiometer at Cerro Paranal (Kerber et al., 2012). In the case of unresolved multiplets, the effective absorption was derived by means of component weights depending on Einstein-$A$ coefficients from HITRAN or NIST. As a final step of the preparation of the time series, the whole period was divided into intervals with a length of 30 min. All data with the middle of the exposure in such an interval were averaged with the individual exposure time as weight. The goal of this procedure was the reduction of the impact of the large

variations in the exposure time. For good data quality in the entire time series, a minimum summed exposure time of 10 min was required for each selected interval. As a consequence, intensity uncertainties are mainly determined by systematic effects such as the quality of the separation of line and continuum, the possible contamination by other airglow lines or remnants of the astronomical targets, the quality of the extinction correction, and uncertainties in the flux calibration.

The procedure for the calculation of climatologies based on the binned time series is described in Sect. 6.1. In this section,

we only focus on the nocturnal annual mean intensity for a reference solar radio flux of 100 sfu, which is derived from the climatological grid data weighted by the nighttime contribution. The climatology-based reference intensities as shown in Figs. 2 and 3 differ from the mean values of the binned time series by less than 10 %. For most lines from the mesopause





region, the deviation is even of the order of only 1 %. The relatively small deviations are related to the fact that the mean solar radio flux of the different time series is already close to 100 sfu.

The UVES-based data sets were also adapted to minimise systematic deviations from the X-shooter data. For the K 770 nm time series from Noll et al. (2019), we added the small atmospheric scattering correction and performed the binning of the data based on 30 min intervals. Despite the different approach to calculate climatologies (see Sect. 6.1), the reference intensity did not significantly change. There are larger changes with respect to the line catalogue of Hanuschik (2003) and Cosby et al. (2006), which was already used for the ESO Sky Model. While Noll et al. (2012) included each emission feature measured by

Hanuschik (2003) as a single entry in the resulting line list, we used the individual components of each feature as derived by Cosby et al. (2006) from theoretical line position and intensity estimates. The latter allowed us to use relatively accurate model wavelengths for each line in the blend and to determine weighting factors for the contribution of the calculated lines to the feature intensity, which was measured by Hanuschik (2003) based on fits of Gaussians. The full line list contains 4,167 unique entries with identified origin. The line intensities were corrected for the van Rhijn effect and atmospheric extinction with the

same recipes as for the X-shooter data and also taking the data from Table 2. In order to derive the relevant zenith angle for the composite spectrum of each set-up, we used the mean airmass values from Hanuschik (2003). For the pwv value, we just set the reference value of 2.5 mm (Table 1). This is not an issue for our analysis as absorption by water vapour is not significant for most crucial lines in the list. We also considered that the flux calibration of Hanuschik (2003) was obviously performed with an outdated extinction curve as the curve of Patat et al. (2011) was not available yet. Finally, for the combined use of

X-shooter and UVES data, we performed additional scaling operations in order to remove remaining differences in the data sets. This was also relevant for the OH data from Noll et al. (2020). Details are given in subsequent subsections which discuss the line intensity models for the different chemical species.

### 4.1 Hydroxyl

The OH radical is the most important contributor to the Earth's nightglow spectrum with the strongest ro-vibrational bands

in the near-IR (Fig. 1). The reference effective emission height is at about 87 km with an FWHM of about 8 km (e.g., Baker and Stair, 1988), although the effective height can vary depending on the line and wave-driven perturbations of the layer by amounts similar to the FWHM in extreme cases (Noll et al., 2022). Vibrationally excited OH in the electronic ground state is mostly produced by the reaction

$$H + O_3 \rightarrow OH^* + O_2 \qquad\qquad\qquad (R1)$$

(Bates and Nicolet, 1950), which preferentially populates the vibrational levels $v = 9$ to 7 with decreasing fractions (e.g., Charters et al., 1971; Llewellyn and Long, 1978; Adler-Golden, 1997). Subsequent photon emissions and collisions with other atmospheric constituents (O, $O_2$, and $N_2$) then redistribute the level populations and finally lead to a predominance of low $v$ (e.g., Dodd et al., 1994; Cosby and Slanger, 2007; Noll et al., 2015). Although the exothermicity of Reaction (R1) is not sufficient, very faint lines with an upper vibrational level of $v' = 10$ were detected (Osterbrock et al., 1998; Cosby and Slanger,

2007), which requires sufficient kinetic energy of the reactants and/or excited $O_3$. As a consequence, the OH emission spectrum



in Fig. 1 contains various ro-vibrational bands with $v'$ between 2 and 10 and lower vibrational levels $v''$ between 0 and 8. Bands with $\Delta v = v' - v'' = 1$ are outside the X-shooter range. The central wavelengths of OH bands increase with decreasing $\Delta v$ and increasing $v'$ (e.g., Osterbrock et al., 1996; Rousselot et al., 2000; Noll et al., 2015). The strongest emissions are related to $\Delta v = 2$.

Each ro-vibrational band consists of R, Q, and P branches (sorted by wavelength). They are characterised by a change of the upper rotational level $N'$ to the lower rotational level $N''$ by $-1$, $0$, and, $+1$, respectively. As the lowest $N$ is defined as 1 for OH (e.g., Pendleton et al., 1993; Dodd et al., 1994; Rousselot et al., 2000), $N' \geq 2$ for R branches and $N'' \geq 2$ for P branches. An example for an OH ro-vibrational band is (9-7) at wavelengths longer than about 2.1 μm in Fig. 1, which shows that P-branch lines are best measurable due to their wider separations. In fact, each branch is divided into two groups related

to the two electronic substates $X^2\Pi_{3/2}$ ($F = 1$) and $X^2\Pi_{1/2}$ ($F = 2$). Lines with $F' = 1$ are stronger than those with $F' = 2$, which is illustrated by the $P_1$ and $P_2$ branches of (9-7). Note that $F'' = F'$ for the visible lines since the intercombination lines are fainter by several orders of magnitude. Lines with high $N'$ are also relatively faint. The $N' = 7$ emissions near 2.30 μm are hard to recognise, whereas $P_1(N' = 2)$ is the strongest P-branch line in the reference spectrum. Finally, each line characterised by the quantum numbers $v$, $N$, and $F$ is actually a doublet. However, the separation of these $\Lambda$ doublets is usually too small

for X-shooter.

The modelling of the complex OH emission pattern can be simplified by the change of the focus from line intensities $I_{i'i''}$ for state transitions from $i'$ to $i''$ to level populations for the upper states $n_{i'}$, which can be derived by

$$n_{i'} = \frac{I_{i'i''}}{A_{i'i''} g_{i'}} \tag{9}$$

(e.g., Noll et al., 2020). This transformation requires the knowledge of the corresponding Einstein coefficients $A_{i'i''}$ in inverse

seconds and the number of degenerate substates contributing to $i'$, i.e. the statistical weight

$$g_{i'} \equiv g' = 4\,(N' - F' + 2) = 2\,(2J' + 1), \tag{10}$$

where $J'$ is the quantum number of the total angular momentum of $i'$. If $\Lambda$ doublets are not separated, this can be implemented by doubling $g'$ and averaging the $A$ coefficients of both components (which are often very similar) under consideration of possible temperature-dependent population differences.

The quality of the level populations resulting from Eq. (9) depends on uncertainties in the $A$ coefficients. There is a long history of theoretical calculations (e.g., Mies, 1974; Langhoff et al., 1986; Goldman et al., 1998; van der Loo and Groenenboom, 2007; Brooke et al., 2016), which showed how challenging the task is for OH. Noll et al. (2020) studied the quality of different sets of coefficients by comparing populations from lines with the same upper level. Based on 544 reliable (partly resolved) $\Lambda$ doublets measured in a UVES composite spectrum (see Sect. 3), the analysis revealed clear systematic deviations, especially

for Q branches and $\Lambda$ doublet components. In order to improve the quality of the populations, Noll et al. (2020) corrected the most recent $A$ coefficients of Brooke et al. (2016) based on the results of the population comparisons with respect to bands, branches, and $\Lambda$ doublets. As Brooke et al. (2016) is included in the HITRAN2020 database (Gordon et al., 2022), the resulting recipes of Noll et al. (2020) are also relevant for our population modelling. However, the UVES-based data do not cover the







**Figure 4.** Correction factors for OH Einstein-$A$ coefficients from HITRAN2020 (Gordon et al., 2022). Each OH band is marked by the wavelength of the $Q_1(1)$ line. Moreover, the upper two panels distinguish between the P (black), Q (green), and R (magenta) branches. In these cases, the wavelength-dependent correction functions originate from the UVES-based fits of Noll et al. (2020) and the values for the X-shooter NIR-arm data of this study. The wavelength range from 0.85 to $1.05\,\mu m$ represents a smooth transition between both regimes. In **(c)**, the band-specific correction factors and their uncertainties are either based on X-shooter data (black circles) or UVES data from Cosby et al. (2006) (green diamonds). Depending on the upper vibrational level, they are given relative to the reference bands marked by blue boxes. OH bands without data for the derivation of the correction are displayed by red crosses. The total correction factor for an OH $\Lambda$ doublet is the product of the $A/A_{\mathrm{HITRAN}}$ of the three panels and depends on the band, branch, and upper rotational level $N'$. The latter needs to be multiplied by the selected $A/A_{\mathrm{HITRAN}}$ in **(a)**.

OH bands in the X-shooter NIR arm. Therefore, we had to extend and revise the correction of the coefficients of Brooke et al.
(2016).





The X-shooter-based sample of OH climatological mean intensities (see Fig. 3) comprises 219 $\Lambda$ doublets in the VIS arm and 325 double lines in the NIR arm. The latter were already measured by Noll et al. (2023b). We used the NIR data for population comparisons between the R and P as well as the Q and P branches. The number of suitable pairs with the same upper level was 60 and 37, respectively. The relatively small numbers are partly explained by the underrepresentation of R- and Q-branch lines in the full sample (about 40 %) due to more severe line blending compared to P-branch lines. Similar to Noll et al. (2020), we performed a linear regression for both branch comparisons with respect to the change of the logarithmic population difference $\Delta y$ as a function of the upper rotational level $N'$. The latter was in maximum 8 for R vs. P and 3 for Q vs. P. The resulting robust slopes and intercepts were the basis for the correction of the $A$ coefficients. For this, we assumed in agreement with Noll et al. (2020) that $\ln(A/A_{\mathrm{HITRAN}})$ corresponds to $-0.5\,\Delta y_{\mathrm{R-P}}$ for P, $\Delta y_{\mathrm{Q-P}} - 0.5\,\Delta y_{\mathrm{R-P}}$ for Q, and $+0.5\,\Delta y_{\mathrm{R-P}}$ for R. The correction of the $A$ coefficients for the OH bands in the NIR arm parameterised by slope and intercept is shown in Figs. 4a and 4b as constant lines for the three branches. While the corrections for the P and R branches are very small, the $A$ coefficients for the Q branch are significantly modified. Note that the values in Fig. 4a need to be multiplied by $N'$ and then added to the values in Fig. 4b to obtain the total branch effect.

In the X-shooter VIS-arm range, we can rely on the UVES-based fits of Noll et al. (2020). Thanks to the very high resolving power of UVES, larger data sets can be used for the comparison of the branch populations. In fact, 101 pairs for R vs. P and 67 pairs for Q vs. P were sufficient to even perform a linear regression analysis for 12 individual OH bands. As the regression slopes and intercepts indicated changes depending on the central wavelength of the band (defined by $\mathrm{Q}_1(N' = 1)$), a wavelength-based second regression analysis was carried out by Noll et al. (2020). The resulting tilted fit lines are also plotted in Figs. 4a and 4b. Such a clear wavelength dependence (especially for the Q branch) was not seen in the NIR-arm data. Hence, we prefer the band-independent but more robust fit there. Moreover, the tilted lines from the UVES-based fit model get close to the constants from the X-shooter-based fits for the reddest bands covered by UVES. In order to remove the remaining discrepancies, we applied a gradual linear transition between both models in the wavelength range from 0.85 to 1.05 μm (marked in the figure), which caused changes in the $A$ coefficient corrections for the (7-3), (8-4), and (3-0) bands with central wavelengths below 1 μm that were fitted by Noll et al. (2020). Except for the intercept for the Q branch, where a local minimum with a depth of a few per cent is produced, the results are convincing.

Another kind of correction is possible with respect to the population differences for bands with the same upper vibrational level $v'$ but different lower vibrational levels $v''$. For this correction, we jointly used the X-shooter VIS- and NIR-arm data in order to obtain a consistent model for a wide wavelength range. As reference bands for each $v'$, we selected the strong ones with $\Delta v = 2$, which should also be more reliable with respect to the theoretical $A$ coefficients than bands with higher $\Delta v$. The only exception is our reference band for $v' = 7$, where we took (7-4) instead of (7-5) because of the strong water vapour absorption, which only allowed us to use two lines. In principle, we could also consider the results from Noll et al. (2020) for this comparison. However, the narrower wavelength range of UVES without access to the $\Delta v = 2$ bands and the fact that all bands measured by Noll et al. (2020) were also covered by the X-shooter data (although with a lower number of selected lines per band on average: about 7 vs. about 11), let us prefer the X-shooter measurements. For each band, we only considered the relatively strong lines with $N' \leq 4$ for the P and R branches plus the $\mathrm{Q}_1(1)$ line in maximum. As the corrections of the





systematic effects related to branch and $N'$ were already performed before, differences in the selected line samples for the individual bands should not have a significant impact.

The results for the band-specific correction of the $A$ coefficients are shown in Fig. 4c. For the X-shooter-related $\ln(A/A_{\text{HITRAN}})$, there is a complex pattern with increasing discrepancies towards shorter wavelengths. Most corrections are positive with a max-
imum of $+0.13$ for (4-0), whereas only (9-3) ($-0.08$) and (7-2) ($-0.11$) are clearly negative. It is not easy to understand the data pattern but the uncertainties of the mean values are relatively small. Moreover, we compared our correction factors to the UVES-based results and found a very good agreement for non-reference bands below $1\,\mu\text{m}$ with differences of the order of $1\,\%$. For the reference bands of Noll et al. (2020), we discovered unexpected non-zero offsets of $-0.16$ for (4-0) and (5-1) and shifts of $0.00$ to $+0.06$ for (9-4), (6-2), (7-3), and (8-3), which point to issues with the data release. Consequently, the modified
$A$ coefficients of the new analysis should be used in any case.

The already discussed UVES-based line catalogue of Cosby et al. (2006) that is based on measurements of Hanuschik (2003) includes 756 OH $\Lambda$ doublets (40 % of them resolved) and 136 cases where only one doublet component was measured. As this line list contains OH bands that were too weak for X-shooter and were not measured by Noll et al. (2020), it is possible to extend the population analysis. First, we had to level out possible intensity differences compared to our X-shooter data due
to differences in the sample properties and the flux calibration. For this, we compared the lines in the catalogue to those in the X-shooter-based line list in order to obtain specific mean corrections for the UVES set-ups centred on 580 and 860 nm that were considered by Cosby et al. (2006). As the spectra of the latter (which show a gap in the centre due to the use of two detector chips denoted by L and U) were taken at different times and the systematic uncertainties in the flux calibration may differ, this division of the wavelength range is reasonable. We found that the UVES-based intensities were higher by
similar mean factors of $1.19$ (860U, $0.86$ to $1.04\,\mu\text{m}$), $1.15$ (860L, $0.67$ to $0.86\,\mu\text{m}$), and $1.18$ (580U, $0.58$ to $0.68\,\mu\text{m}$) with uncertainties of about $0.02$. The main reason for the deviation from $1.0$ is probably the relatively high solar activity in 2001 when the UVES data were taken (Sect. 3). We corrected the discrepancies using the given factors, also considering the value for 580U for 580L ($0.48$ to $0.58\,\mu\text{m}$). Then, we also calculated band-specific population differences for the Cosby et al. (2006) data. Overall, the pattern is similar to the X-shooter-based results but with some scatter. Most deviations were smaller than $0.05$
but cases up to $0.11$ were also found. As the 580L range also covers four OH bands not measured in the X-shooter spectra, we were able to extend the correction of the $A$ coefficients in Fig. 4c. While the $\ln(A/A_{\text{HITRAN}})$ of (9-2) and (7-1) are strongly positive but with low uncertainties, the values of the very faint bands (8-1) and (6-0) with line intensities much smaller than $1\,\text{R}$ are negative with high uncertainties. For the remaining bands without data, we kept $\ln(A/A_{\text{HITRAN}})$ at $0$ as in the case of the reference bands.

Noll et al. (2020) also derived corrections for the ratio of the $A$ coefficients of the $\Lambda$ doublet components, which is usually considered to be unity in theoretical calculations. However, especially doublets with high $N'$ can show significant deviations. This correction does not affect the subsequent population analysis, which is only based on combined $\Lambda$ doublets, but was relevant for the calculation of the OH line intensities based on the HITRAN database for the PALACE line model.

With the corrected $A$ coefficients, the populations resulting from Eq. (9) are more consistent. The natural logarithm of
the selected 544 X-shooter-based populations in rayleigh seconds, $y$, is plotted in Fig. 5 as a function of the level energy $E$





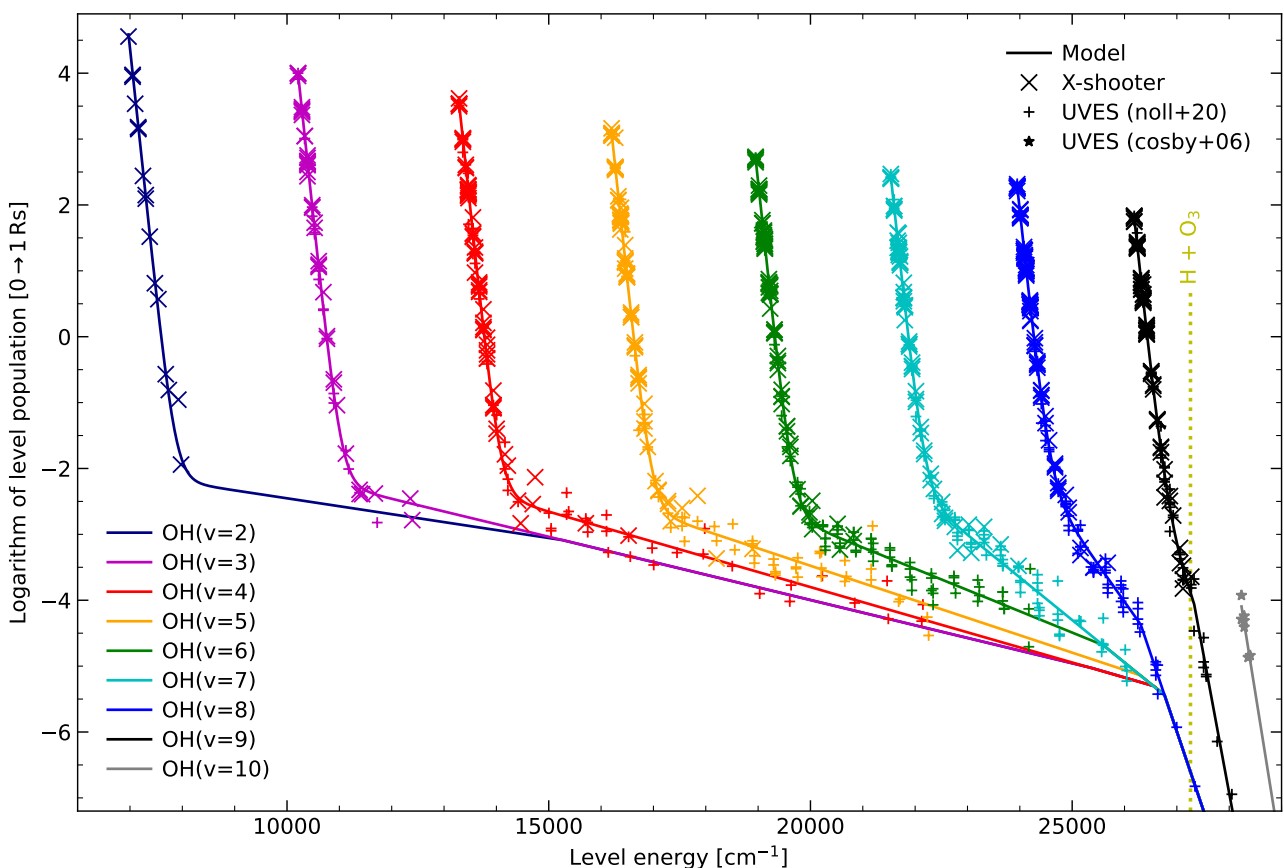

**Figure 5.** OH population model for the electronic ground state and vibrational levels $v = 2$ to 10 in logarithmic units and depending on the level energy. For $v \leq 7$, 8 to 9, and 10, the model (solid lines) consists of two, three, and one fitted components, respectively. The corresponding parameters are provided in Table 3. The basic data for the $v \leq 9$ model originate from X-shooter measurements (crosses) and the UVES-related data from Noll et al. (2020) (plus signs). For $v = 10$, UVES-based data from Cosby et al. (2006) (stars) were used. The energy provided by the OH-producing reaction of H and $O_3$ is marked by a vertical dotted line.

in inverse centimetres. As the UVES-based line list of Noll et al. (2020) also contains relatively faint lines with higher $N'$ (7.2 vs. 3.9 on average), we also included the corresponding 664 populations in our analysis in order to achieve more complete population distributions. For this purpose, we had to correct possible systematic discrepancies caused by the sample differences and the flux calibration. From the comparison of 135 lines, we found that the UVES-based populations had to be increased by
$\Delta y$ of 0.096 for vibrational level $v = 3$, 0.063 for $v = 4$, 0.042 for $v = 5$, and between 0.030 and 0.034 for the higher $v$. The corrected populations are also plotted in Fig. 5. There are multiple population measurements for most levels depending on the number of measured lines with different lower levels and the coverage of a level by the two samples. Finally, we also used the previously corrected data from Cosby et al. (2006). Here, we only focused on lines with $v' = 10$, which are not present in the two other catalogues. The list of safe detections is short. It only contains one (10-4) and six (10-5) $\Lambda$ doublets. Hence, possible





**Table 3.** Fit parameters and their uncertainties for cold and hot OH and $O_2$ vibrational level populations

| Mol. | $e$ | $v$ | Data[a] | $N_{\mathrm{sel}}$[b] | $E_0$[c] cm$^{-1}$ | $y_{0,\mathrm{cold}}$[d] ln(R s) | | $T_{\mathrm{cold}}$[e] K | | $y_{0,\mathrm{hot}}$[d] ln(R s) | | $T_{\mathrm{hot}}$[e] K | |
|---|---|---|---|---|---|---|---|---|---|---|---|---|---|
| OH | X | 2 | xuo | 18 | 6,971.37 | 4.591 | 0.108 | 191.0 | fixed | −2.090 | fixed | 12,000. | fixed |
| OH | X | 3 | xuo | 75 | 10,210.57 | 3.994 | 0.022 | 191.0 | fixed | −2.118 | fixed | 7,500. | fixed |
| OH | X | 4 | xu | 155 | 13,287.18 | 3.538 | 0.026 | 191.0 | fixed | −2.265 | 0.045 | 6,294. | 285. |
| OH | X | 5 | xu | 166 | 16,201.32 | 3.088 | 0.027 | 191.0 | fixed | −2.475 | 0.055 | 5,455. | 341. |
| OH | X | 6 | xu | 210 | 18,951.87 | 2.701 | 0.019 | 191.0 | fixed | −2.540 | 0.032 | 4,469. | 158. |
| OH | X | 7 | xu | 185 | 21,536.21 | 2.415 | 0.023 | 191.0 | fixed | −2.067 | 0.035 | 2,233. | 48. |
| OH | X | 8 | xu | 219 | 23,949.83 | 2.262 | 0.015 | 191.0 | fixed | −1.732 | 0.048 | 1,288. | 35. |
| | | | xu | 13 | 26,309.32 | | | | | −4.369 | 0.062 | 611. | 33. |
| OH | X | 9 | xu | 165 | 26,185.80 | 1.780 | 0.016 | 191.0 | fixed | −1.937 | 0.286 | 760. | 108. |
| | | | xu | 12 | 27,321.20 | | | | | −4.020 | 0.105 | 336. | 24. |
| OH | X | 10 | c | 7 | 28,234.08 | | | | | −4.090 | 0.085 | 290. | 58. |
| $O_2$ | a | 0 | x | 8 | 7,892.02 | 15.747 | 0.015 | 193.1 | 1.4 | | | | |
| $O_2$ | a | 1 | x | 0 | 9,375.37 | 4.693 | fixed | 193.1 | fixed | | | | |
| $O_2$ | b | 0 | x | 8 | 13,122.01 | 7.359 | 0.010 | 187.3 | 1.2 | | | | |
| | | | c | 30 | 13,122.01 | (7.572) | (0.045) | (195.0) | (2.9) | −1.963 | 0.519 | 2,167. | 759. |
| $O_2$ | b | 1 | c | 44 | 14,526.74 | 0.406 | 0.137 | (195.0) | fixed | −1.679 | 0.504 | 6,181. | high |
| $O_2$ | b | 2 | c | 24 | 15,903.50 | 1.100 | high | (195.0) | fixed | −1.807 | high | 6,181. | fixed |

[a] x = X-shooter data (NIR-arm OH intensities already used by Noll et al. (2023b)), u = UVES data from Noll et al. (2020), c = UVES data from Cosby et al. (2006) ($O_2$-related fit values in parentheses replaced by X-shooter-based results for the final model), o = fit parameters for OH low-$v$ hot populations from Oliva et al. (2015).

[b] Number of line measurements used for the fit.

[c] Minimum level energy according to HITRAN2020 (Gordon et al., 2022).

[d] Logarithm of population of individual level at $E_0$.

[e] Effective (pseudo-)temperature of population distribution.

Uncertainties: values given for uncertainties below 100 % (otherwise 'high'), 'fixed' for parameters without a fit.

differences in the HITRAN-related $A$ coefficients for these two bands could not be corrected (see Fig. 4c). The resulting populations are also displayed in Fig. 5.

The distribution of the plotted level populations shows a steep decrease for low rotational levels $N$ for each $v$. For high $N$ (if available), the drop is distinctly weaker, especially for low $v$. Moreover, the populations tend to decrease with increasing





$v$ with a major decrease between $v = 9$ and 10, which is obviously related to the exothermicity limit of Reaction (R1) of
about $27,300\,\mathrm{cm}^{-1}$. The structure of the population pattern is well known (e.g., Pendleton et al., 1993; Cosby and Slanger,
2007; Oliva et al., 2015; Noll et al., 2015, 2020). For its characterisation, it can be exploited that the $v$-specific logarithmic
populations for low and high $N$ are located along relatively straight lines. The slopes $\frac{\mathrm{d}y}{\mathrm{d}E}$ of these lines can be converted into
pseudo-temperatures $T$ by

$$T = -\frac{1}{k_\mathrm{B}\frac{\mathrm{d}y}{\mathrm{d}E}} \tag{11}$$

(Mies, 1974; Noll et al., 2018), where $k_\mathrm{B}$ is the Boltzmann constant. The steep population decrease for low $N$ is therefore
related to low $T$, which should be close to the ambient temperature in the emission layer, whereas high-$N$ populations are
characterised by high pseudo-temperatures, which reflect a lack of thermalisation of the nascent population distribution from
Reaction (R1) by collisional processes (e.g., Pendleton et al., 1989, 1993; Dodd et al., 1994; Cosby and Slanger, 2007; Kaloger-
akis et al., 2018; Noll et al., 2018). This bimodal distribution for fixed $v$ can be best fitted by a two-temperature model (Oliva
et al., 2015; Kalogerakis et al., 2018; Kalogerakis, 2019a; Noll et al., 2020). We therefore fitted the populations for $v$ from 2 to
9 using

$$n(E) = n_{0,\mathrm{cold}}e^{-(E-E_0)/(k_\mathrm{B}T_\mathrm{cold})} + n_{0,\mathrm{hot}}e^{-(E-E_0)/(k_\mathrm{B}T_\mathrm{hot})}, \tag{12}$$

where $n_0$ refers to the population at the lowest energy $E_0$ of a given $v$.

Fits with unconstrained parameters were carried out for $v$ of 4 to 9. The results agree within the uncertainties with the results
of Noll et al. (2020). Hence, the inclusion of the X-shooter data and the slightly modified $A$ coefficients for the UVES data
only had a minor impact on the best fit parameters. However, the increase of the sample size reduced the uncertainties. The fits
showed very similar values for $T_\mathrm{cold}$ with a mean of $191.6 \pm 0.7\,\mathrm{K}$ for $v$ between 5 and 9 and a slight outlier of about $196\,\mathrm{K}$
for $v = 4$. This suggests to fix $T_\mathrm{cold}$ and set it to the ambient temperature. SABER-based temperature profiles for the region
around Cerro Paranal indicate that the mean profile is fairly constant in most of the altitude range relevant for OH with changes
of the order of only $1\,\mathrm{K}$ (Noll et al., 2016). The SABER data set from 2002 to 2015 collected by Noll et al. (2017) indicates a
mean temperature of about $191\,\mathrm{K}$ consistent with the population fits. Hence, we repeated the fits with this value as $T_\mathrm{cold}$. The
resulting fit parameters (with $y_0 = \ln(n_0)$) are shown in Table 3. With increasing $v$, $y_{0,\mathrm{cold}}$ and $T_\mathrm{hot}$ decrease (from about 6,300
to $800\,\mathrm{K}$ for the latter), whereas there is no clear trend for $y_{0,\mathrm{hot}}$. For $v$ of 2 and 3, we needed to set additional constraints as the
hot populations are not sufficiently covered. Fortunately, Oliva et al. (2015) were able to perform two-component fits for these
low $v$, based on line measurements between 0.95 and $2.4\,\mu\mathrm{m}$ in a high-resolution spectrum of the GIANO echelle spectrograph
at the island of La Palma (Spain). For $v = 2$, we directly took their results for the hot population, i.e. $T_\mathrm{hot} = 12,000\,\mathrm{K}$ and a
ratio of $n_{0,\mathrm{hot}}$ and $n_{0,\mathrm{cold}}$ of $0.14\,\%$. For $v = 3$ with at least some data points in the transition region between cold and hot
population, we used the values of Oliva et al. (2015) (7,000 K and 0.23 %) as a starting point for our own fits, which then
showed the most convincing results (also with respect to the higher $v$) for fixed values of 7,500 K and 0.22 %. The changes
should not be larger than the fit uncertainties for the GIANO data (see Kalogerakis et al., 2018).





Close to the exothermicity limit of about $27{,}300\,\mathrm{cm}^{-1}$, the two-component fits do not work as the populations decrease faster than given by $T_{\mathrm{hot}}$. For this reason, the fits of $v = 8$ and 9 in Table 3 were performed with upper energy limits of $26{,}300\,\mathrm{cm}^{-1}$ for $v = 8$ ($N \leq 13$) and $27{,}330\,\mathrm{cm}^{-1}$ for $v = 9$ ($N \leq 9$). For the populations above these limits, fits with a single temperature turned out to be promising. In a plot of $v$ vs. $E$ as Fig. 5, this is just a linear regression. We performed the fits including the

data points with the highest $N$ that were used for the two-component fits, i.e. 13 and 9, respectively. However, we only applied the results for levels with higher $N$. In Table 3, the listed $E_0$ refer to the intersection of the one- and two-component fits. The resulting temperatures are only about half the $T_{\mathrm{hot}}$ from the two-component model but still distinctly higher than $T_{\mathrm{cold}}$. The small sample of $v = 10$ population measurements was also fitted with a one-component model. The resulting temperature is about 300 K. It might be somewhat lower than the corresponding fit for $v = 9$. The $y_0$ are very similar for both $v$.

The complete OH population model is shown in Fig. 5. For very high $N$ not covered by data, we introduced the rule that curves for different $v$ never cross. Instead, the curve of the lower $v$ then follows the path of the next higher one. This recipe considers that the measured high-$N$ populations of the different $v$ tend to converge. This interpretation is also supported by similar effective emission heights (Noll et al., 2022) and variability patterns (Noll et al., 2023b) of high-$N'$ lines. The population model shows good agreement with the measured populations. The mean population ratio is very close to 1.00 and

the standard deviation is 0.10 for the X-shooter and 0.21 for the Noll et al. (2020) UVES data. In the case of levels with $N \leq 4$ neglecting Q-branch lines, the scatter is only 0.05 for both data sets. The main reason for the deviations should be measurement uncertainties, but higher-order population features not covered by the model may also contribute. For the PALACE line list, the OH population model was applied to 11,029 $\Lambda$ doublets from HITRAN2020. Using the corrected $A$ coefficients, the resulting total reference intensity (without atmospheric extinction) amounts to $715\,\mathrm{kR}$ in the full model range from 0.3 to $2.5\,\mu\mathrm{m}$. The

strongest band is (4-2) at about $1.6\,\mu\mathrm{m}$ with $92\,\mathrm{kR}$ for all band-related lines. For $\Delta v = 3$, (8-5) at about $1.3\,\mu\mathrm{m}$ indicates the highest reference intensity of $24\,\mathrm{kR}$. For $\Delta v$ from 4 to 7, where also measured data exist, the bands with $v' = 9$ are the most intense ones showing $3.8\,\mathrm{kR}$, $570\,\mathrm{R}$, $61\,\mathrm{R}$, and $5.1\,\mathrm{R}$ for $v''$ from 5 to 2. The weakest band of this list, i.e. (9-2), emits near $520\,\mathrm{nm}$. The total reference emission of all $v' = 10$ bands amounts to about $330\,\mathrm{R}$. The integrated OH emission between 642 and $858\,\mathrm{nm}$ is about $3.1\,\mathrm{kR}$, which is very close to $3.2\,\mathrm{kR}$ as reported by Noll et al. (2012) for the ESO sky model, although

for a solar radio flux of about $129\,\mathrm{sfu}$.

So far, the discussion has focused on the main isotopologue $^{16}$OH. However, other isotopologues can also emit. Cosby et al. (2006) succeeded to identify 26 $\Lambda$ doublets of $^{18}$OH (covering only one component in five cases). The lines were very faint with intensities below $1\,\mathrm{R}$ and a total emission of $8.4\,\mathrm{R}$. Compared to the model intensities of the same lines for $^{16}$OH, we found a mean ratio of $0.00241 \pm 0.00018$ for the 15 most reliable $\Lambda$ doublets. This result is consistent with $0.00235 \pm 0.00007$

obtained by Osterbrock et al. (1998) based on a different data set. The ratio is higher than the standard abundance ratio of 0.00201 from the HITRAN database. Deviations can be related to differences in the energy levels and $A$ coefficients. The discrepancy may also be caused by an enrichment of heavier O isotopes in $O_3$ (Osterbrock et al., 1998). As HITRAN does not contain $^{18}$OH lines in the PALACE wavelength range, we only included the lines identified by Cosby et al. (2006) in our line model. This is certainly highly incomplete. However, the small number of detected lines also shows that the contributions

of heavier OH isotopologues to the airglow emission spectrum are mostly negligible. For the model, we took the measured



intensities (with the general corrections discussed before) and the modelled wavelengths from Cosby et al. (2006). For $^{16}$OH, the latter are in good agreement with the wavelengths from the HITRAN database. The relative deviations are only of the order of $10^{-7}$.

### 4.2 Molecular oxygen

Excited $O_2$ has a significant impact on the shape of the nocturnal airglow (nightglow) spectrum (see Figs. 1 and 2). In contrast to OH where the emission is only related to ro-vibrational bands of the electronic ground state $X^2\Pi$, the relevant $O_2$ bands always involve electronic transitions including the states $X^3\Sigma_g^-$, $a^1\Delta_g$, $b^1\Sigma_g^+$, $c^1\Sigma_u^-$, $A'^3\Delta_u$, and $A^3\Sigma_u^+$, which are listed with increasing energy. Relevant band systems are (A-X), (A'-a), (c-b), (c-X), (b-X), and (a-X) (e.g., Rousselot et al., 2000; Slanger and Copeland, 2003; Cosby et al., 2006; Noll et al., 2016; von Savigny, 2017). Systems related to the higher electronic levels

cause relatively weak bands in the near-UV to visual wavelength regime. The (b-X)(0-0) band at $762\,\mathrm{nm}$ is very strong but not visible from the ground as the lower level is $X^3\Sigma_g^-(v=0)$, which is strongly affected by atmospheric $O_2$ absorption (see Fig. 1). The (b-X)(0-1) band at $865\,\mathrm{nm}$ is significantly fainter but not affected by self-absorption. There are other but weak bands of this system with $v' \leq 15$. The strongest nightglow band at all is (a-X)(0-0) at $1.27\,\mathrm{\mu m}$. Although also strongly affected by self-absorption, it remains a relatively bright band for observations at the ground. It is also clearly brighter than (a-X)(0-1)

at $1.58\,\mathrm{\mu m}$, the only other known band of the system.

The classical nighttime production mechanism for excited $O_2$ is atomic oxygen recombination

$$O + O + M \rightarrow O_2^* + M \tag{R2}$$

(e.g., Slanger and Copeland, 2003), where M can be any atmospheric constituent. The reaction preferentially produces populations in the higher electronic states. Then, the lower states are essentially populated by level-changing collisions of the excited

molecules. However, there appear to be other important processes. Kalogerakis (2019b) proposes that $b^1\Sigma_g^+$ could be excited up to $v=1$ by collisions with O in the excited $^1$D state, which could be produced by collisions of O in the ground state $^3$P with excited OH. Concerning $a^1\Delta_g$, Noll et al. (2024a) discussed pathways such as reactions of OH and O or reactions involving $HO_2$. Nevertheless, there is no satisfying explanation, so far. In any case, it needs to be explained why the nighttime (a-X) emission peaks several kilometres below the emissions of the other band systems. For (b-X)(0-0), $94\,\mathrm{km}$ is a typical peak

height (e.g., Yee et al., 1997), whereas it is about $90\,\mathrm{km}$ for (a-X)(0-0) as measured at Cerro Paranal using SABER profiles from the second half of the night (Noll et al., 2016). At the beginning of the night, most emission originates from significantly lower heights with contributions even from the lower mesosphere. This change in the emission profile is caused by populations produced by $O_3$ photolysis at daytime, which can produce excited $O_2(a^1\Delta_g)$. As this state has a long effective lifetime of about 1 hour in the mesopause region, this pathway still matters in the first few hours of the night indicated by an exponential

intensity decrease (López-González et al., 1989; Mulligan and Galligan, 1995; Noll et al., 2016). For the van Rhijn correction, we chose $89\,\mathrm{km}$ as a representative centroid height for the major fraction of the night (see Table 2).

In order to model the mean $a^1\Delta_g(v=0)$ population, we focused on the strong (a-X)(0-0) band. As this band consisting of nine branches (e.g., Rousselot et al., 2000) shows high line density especially in the band centre and is strongly affected by



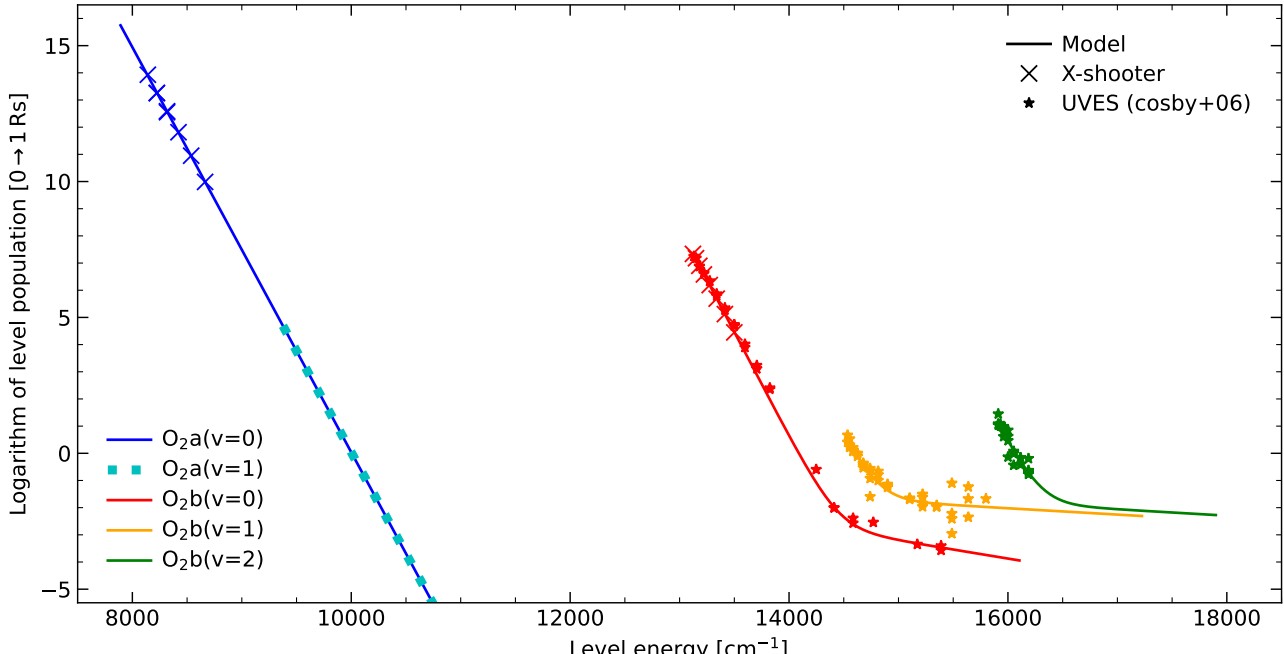

**Figure 6.** $O_2$ population model for the electronic states $a^1\Delta_g$ for vibrational levels $v \leq 1$ and $b^1\Sigma_g^+$ for $v \leq 2$ in logarithmic units and depending on the level energy. The model (solid or dotted lines) consists of a linear fit for both $v$ of $a^1\Delta_g$ or two fitted components for each $v$ of $b^1\Sigma_g^+$. The corresponding parameters are provided in Table 3. The basic data for the $O_2$ population model originate from X-shooter measurements (crosses) and UVES-based data from Cosby et al. (2006) (stars).

self-absorption, only a small number of lines is suitable for measurements at X-shooter resolution. With a small data set of

X-shooter data, such measurements were already performed by Noll et al. (2016). We carried out a similar line selection only considering lines with relatively high $N'$ of the branches $^SR$ and $^OP$, which are related to an $N$ change of $-2$ and $+2$ (denoted by the letters S and O), respectively, and an opposite spin change by 1 leading to $-1$ (R) and $+1$ (P) in terms of the total angular momentum $J$. Our final sample consists of the four $^SR$ lines with $N'$ of 15, 17, 21, and 23 at wavelengths between 1.249 and 1.256 µm and the four $^OP$ lines with $N'$ of 13, 15, 17, and 19 at wavelengths between 1.282 and 1.289 µm (see Fig. 3). These

lines are relatively far from the crowded band centre and only experience moderate self-absorption with zenith transmissions between 0.22 and 0.87, which can be corrected in a reliable way. The reference intensities from the derived climatologies were then converted into populations using Eq. (9) and $O_2$ $A$ coefficients from HITRAN2020 (Gordon et al., 2022). The latter are more robust than in the case of OH and do not need to be optimised. We fitted the data with a single temperature (see Fig. 6) and obtained $193.1 \pm 1.4$ K as shown by Table 3. This is very close to the expected ambient temperature and therefore reliable.

In the view of the mentioned long lifetime of $a^1\Delta_g(v=0)$, the rotational population should be well thermalised. Hence, the resulting $T_{\text{cold}}$ should be representative for all rotational levels. We therefore applied the fit results to all levels in order to





calculate reference intensities for the (0-0) band with a total strength of $155\,\mathrm{kR}$ without self-absorption (otherwise only about $17\,\mathrm{kR}$) and the much weaker (0-1) band with $2.0\,\mathrm{kR}$.

The HITRAN $O_2$ line list also includes the (a-X)(1-0) band near $1.07\,\mu\mathrm{m}$. Noll et al. (2024a) speculated whether a feature in the residual airglow continuum (see Sect. 5) could be related to this band. We tested this by calculating the (a-X) model with different factors for the ratio of the $v = 1$ and $v = 0$ populations of $\mathrm{a}^1\Delta_\mathrm{g}$ and comparing the resulting spectrum to an X-shooter NIR-arm mean spectrum. However, we found that the calculated and measured structures do not match suggesting only a very small contribution of (a-X)(1-0). We therefore assumed that the vibrational temperature equals the rotational temperature (see Fig. 6 and Table 3), which resulted in a band intensity of only $0.012\,\mathrm{R}$. The calculated strength of the also available (1-1) band of $2.6\,\mathrm{R}$ is much higher, but the band is located close to the strong (0-0) emission. For the latter, the HITRAN database also includes 322 lines of the $^{16}\mathrm{O}^{18}\mathrm{O}$ isotopologue. For the scaling of these lines compared to those of $^{16}\mathrm{O}_2$, we applied the standard abundance ratio of 0.0040 from HITRAN, which resulted in a band strength of $1.3\,\mathrm{kR}$. The whole (a-X) model comprises 1,196 lines.

The only sufficiently strong (b-X) band that can be measured with X-shooter is (0-1) at about $865\,\mathrm{nm}$. It is not affected by self-absorption. In agreement with the X-shooter-based measurements of Noll et al. (2016), we focused on the P branch, where the lines are stronger and better separated than in the R branch. Nevertheless, the two subbranches $^\mathrm{P}Q$ and $^\mathrm{P}P$, which are related to the multiplicity of the electronic ground state $\mathrm{X}^3\Sigma_\mathrm{g}^-$, are not separated for the same $N'$. Only in the case of the minimum $N'$ of 0, it is a single line as the $^\mathrm{P}Q$ component is forbidden. We selected this single line and seven doublets with $N'$ from 4 to 16 (odd numbers are not allowed) to study the population of $\mathrm{b}^1\Sigma_\mathrm{g}^+(v = 0)$. A linear fit of the logarithmic populations calculated by means of the climatological mean intensities and the corresponding HITRAN $A$ coefficients returned $187.3 \pm 1.2\,\mathrm{K}$ as convincing single temperature (see Table 3 and Fig. 6). As the (b-X)(0-1) emission heights are closer to the temperature minimum of the mesopause, the resulting $T_\mathrm{cold}$ should be slightly lower than in the case of (a-X)(0-0) and the different OH bands (Noll et al., 2016).

The fit model works very well for the covered range of $N'$. However, there might be discrepancies for distinctly higher rotational levels. The lifetime of $\mathrm{b}^1\Sigma_\mathrm{g}^+(v = 0)$ is only about $11\,\mathrm{s}$ (e.g., Zhu and Kaufmann, 2019) (compared to more than an hour for $\mathrm{a}^1\Delta_\mathrm{g}$) and the chemistry also seems to differ as discussed above. Fortunately, we could investigate this by using the UVES-based line catalogue of Cosby et al. (2006), where we selected 21 resolved (b-X)(0-1) P-branch lines with $N'$ between 2 and 22 (0 had calibration issues) and nine (b-X)(0-0) P-branch lines with $N'$ between 28 and 40. The latter were measurable as self-absorption becomes weaker with increasing $N'$. The reference transmission values were between 0.15 and 1.00 for the selected lines. Before the study of the populations, we had to scale the intensities to be consistent with the X-shooter data. For this, we compared (b-X)(0-1) lines which are present in both data sets. The logarithmic line ratios were linearly fitted depending on $N'$ in order to obtain a robust intensity ratio for $N' = 0$. The result was $1.55 \pm 0.03$, i.e. the Cosby et al. (2006) data were significantly brighter. The factor is most probably caused by the high solar activity for the UVES data. In June 2001, the monthly mean was $174\,\mathrm{sfu}$. As discussed in Sect. 6.2, $O_2$ lines show a relatively strong correlation with the solar radio flux.

After the correction of the lines intensities, which we applied to all $O_2$ lines of Cosby et al. (2006), we calculated the level populations using the HITRAN database and then fitted the population distribution using the two-component model in Eq. (12).

 

As indicated by Fig. 6 and Table 3, the fit revealed a weak hot population with roughly $2,200 \pm 800\,\mathrm{K}$ that is only visible in the highest $N$. Moreover, $T_{\mathrm{cold}}$ resulted in $195.0 \pm 2.9\,\mathrm{K}$, which is distinctly higher than in the case of the X-shooter data. The most likely explanation is also the high solar activity during the UVES observations. As the reference intensities should
be valid for a solar radio flux of $100\,\mathrm{sfu}$, we constructed our final $\mathrm{b}^1\Sigma_{\mathrm{g}}^+(v=0)$ model by using the X-shooter-based fit for the cold population and only the hot population from the UVES-based fit. For this approach, it was important to have a scaling factor for the UVES-based intensities optimised for $N'=0$ as described above. The resulting reference intensity for the whole (b-X)(0-0) band (150 lines) is $6.5\,\mathrm{kR}$ without self-absorption, which is in good agreement with satellite-based measurements (e.g., Yee et al., 1997). The corresponding intensity for (b-X)(0-1) amounts to $340\,\mathrm{R}$. The 49 lines for this band include two
$N'=24$ lines, where the intensities (about $0.5\,\mathrm{R}$ in sum) were directly taken from the corrected Cosby et al. (2006) data set since these lines are not included in HITRAN2020. The derived reference band intensity is very close to the one for the ESO Sky Model of $320\,\mathrm{R}$ (Noll et al., 2012). However, the latter would be only about $275\,\mathrm{R}$ for a solar radio flux of 100 instead of $129\,\mathrm{sfu}$.

Slanger et al. (2000) discussed the detection of various (b-X) bands with $v'$ between 1 and 15 in spectra of the High
Resolution Echelle Spectrometer (HIRES) from Mauna Kea (Hawai'i). We could also extract more than 2,000 of such lines from the line list of Cosby et al. (2006). The HITRAN database only covers a small number of these bands, essentially only those that matter for absorption calculations such as (1-0), (1-1), (2-0), and (2-1). Hence, our approach was to directly use the calculated wavelengths (with a tested systematic uncertainty significantly below $10^{-6}$) and the scaled intensities from Cosby et al. (2006) as primary input for the model. However, the latter can have relatively large measurement uncertainties, especially
for the relatively faint $v' \geq 1$ lines. Moreover, the catalogue is incomplete due to line blending and detection limits, and there is a possible issue with respect to the high solar radio flux during the UVES observations as discussed before. Therefore, we performed population fits for lines with UVES and HITRAN data in order to improve the model, although the covered $v'=1$ and 2 bands appear to be relatively faint compared to the integrated emission for higher $v'$ such as 3, 4, 5, and even 12 (Slanger et al., 2000).

Using 44 (1-1) lines, a two-component fit (see Fig. 6) with $T_{\mathrm{cold}} = 195.0\,\mathrm{K}$ from the (0-1) fit resulted in a hot population with an uncertain temperature of about $6,000\,\mathrm{K}$ (see Table 3). For the fit of 24 (2-1) lines, $T_{\mathrm{hot}}$ was also fixed (using the previous result) as the measured lines do not cover the hot population (Fig. 6). In a similar way as for the $\mathrm{b}^1\Sigma_{\mathrm{g}}^+(v=0)$ model, we then replaced $T_{\mathrm{cold}}$ in both $v'$ cases by $187.3\,\mathrm{K}$ from the X-shooter-based fit of the (0-1) band. The resulting summed reference intensity for the 272 lines from the HITRAN database with $v'=1$ and 2 and $v'' \leq 1$ amounts to $14\,\mathrm{R}$. For the remaining $v' \geq 1$
bands, the integrated intensity from 460 measured lines results in $83\,\mathrm{R}$. With additional three weak lines from the (0-2) band, the total line number of the (b-X) model for $^{16}\mathrm{O}_2$ is 934. These lines emit $6.9\,\mathrm{kR}$ under reference conditions.

HITRAN2020 also includes lines related to $\mathrm{b}^1\Sigma_{\mathrm{g}}^+(v=0)$ for the $^{16}\mathrm{O}^{18}\mathrm{O}$ isotopologue. As the UVES-based data set also contains such measurements, we could scale the population model for $v=0$ by means of a comparison of level populations from the measured intensities of (0-0) for $^{16}\mathrm{O}^{18}\mathrm{O}$ and (0-1) for $^{16}\mathrm{O}_2$. The resulting reference intensity model for the former
comprises 140 lines with a summed emission of $20\,\mathrm{R}$, which corresponds to 0.0031 times the radiation from the (0-0) band of $^{16}\mathrm{O}_2$. This is lower than the standard abundance ratio of 0.0040 but consistent with Slanger et al. (2000) based on the HIRES





data set. HITRAN also contains the (1-0) and (2-0) bands for $^{16}O^{18}O$, which could also be modelled using the corresponding population fits. However, the 216 very weak lines have a negligible impact on the total emission for $^{16}O^{18}O$. For the (b-X) model, we also considered 13 detected lines of the (0-0) band of $^{16}O^{17}O$ from the Cosby et al. (2006) data set without
population calculations. The total emission (which represents only a minor fraction of the lines of this band) is just $1.3\,R$.

Lines of the high-energy $c^1\Sigma_u^-$, $A'^3\Delta_u$, and $A^3\Sigma_u^+$ states are not part of HITRAN2020 in the PALACE model range. Therefore, we directly used 1,590 corresponding lines in the wavelength range from 314 to $550\,nm$ identified by Cosby et al. (2006). We also applied the intensity scaling factor of $1/1.55$ as derived for (b-X)(0-1) and $N' = 0$. However, the difference in the states and the UVES set-ups (mainly 346 and 437 instead of 860; Hanuschik, 2003) could cause a different factor. Therefore,
we compared the integrated intensities of several clear emission features in the range from 330 to $400\,nm$ (see Fig. 1) as derived from a preliminary model and an X-shooter UVB-arm mean spectrum, which revealed that our model was still too bright by a factor of about $1.1 \pm 0.1$. We therefore divided the intensities by 1.1 to obtain our final model for the high-energy states. The total reference emission amounts to about $62\,R$. The contributions of the (A-X), (A'-a), (c-b), and (c-X) band systems are 35.7, 22.6, 3.4, and $0.3\,R$, respectively. As the bands of the different systems strongly overlap and the individual lines are very weak
($< 0.3\,R$), the line list is highly incomplete. The lack of coverage is particularly severe for intensities below about $0.01\,R$ as Fig. 2 demonstrates. Therefore, the PALACE model of the $O_2$ emission related to the $c^1\Sigma_u^-$, $A'^3\Delta_u$, and $A^3\Sigma_u^+$ states also consists of an unresolved continuum component, which is discussed in Sect. 5.3.

### 4.3 Sodium

The alkali metal $Na$ produces a prominent doublet at air wavelengths of 589.0 and $589.6\,nm$ (see Fig. 1). Accurate wavelengths
can be taken from the NIST atomic line database (Kramida et al., 2023). These lines are usually called $D_2$ and $D_1$ and are related to the transitions from the excited $^2P_{3/2}$ and $^2P_{1/2}$ levels to the ground state $^2S_{1/2}$. The excitation mechanism is a multi-step process starting with

$$Na + O_3 \rightarrow NaO^* + O_2 \tag{R3}$$

and ending with

$$NaO^{(*)} + O \rightarrow Na^* + O_2. \tag{R4}$$

The version of Reaction (R4) with $NaO$ in excitation (marked by an asterisk) is the classical mechanism of Chapman (1939). In addition, laboratory measurements by Slanger et al. (2005) indicated that $NaO^*$ from Reaction (R3) can be first deactivated by collisions, most likely with $O_2$, before Reaction (R4) starts. For both processes, it is also possible that $Na$ is produced in the ground state, which does not generate airglow emission. For the effective emission height, we take $92\,km$, which agrees well
with satellite-based measurements (e.g., Koch et al., 2022) and is consistent with Noll et al. (2012).

Both lines can be well measured in X-shooter VIS-arm spectra. However, the $D_2$ line is also used at Cerro Paranal for the production of artifical guide stars via laser-induced fluorescence in the $Na$ layer. The amount of contamination depends on the angular distance of the line of sights of the two telescopes operating X-shooter and the laser if it is used. Hence, we had to





perform a very careful outlier detection in terms of the $D_2$ intensity and the $D_2$-to-$D_1$ ratio. Strong outliers are easily found but

the identification of small contributions remains uncertain. Due to the different pathways for the Na excitation, the $D_2$-to-$D_1$ ratio is variable (Slanger et al., 2005). For a subsample consisting of the most reliable measurements, we found a mean ratio of 1.73. This is well in the range of values reported by Slanger et al. (2005) based on spectra from Mauna Kea. For the line list of Cosby et al. (2006), about 1.70 was obtained. For the calculation of the $D_1$ climatology, we used a 4 times larger sample (27,942 vs. 6,999). As the same for the $D_2$ line would have caused contaminations by the laser, we derived the reference intensity for

$D_2$ by just multiplying the result of 13.4 R for $D_1$ by the factor 1.73, which returned 23.1 R. Both intensities are plotted in Figs. 2 and 3. The resulting total Na emission amounts to 36.5 R. This is slightly lower than the corresponding value of 43 R in the ESO Sky Model. Unterguggenberger et al. (2017) obtained from a sample of 3,662 X-shooter spectra (only covering the period up to March 2013) about 40 R.

### 4.4 Potassium

The airglow emission of the alkali metal K is also characterised by a D doublet with the same upper and lower states as in the case of Na (Sect. 4.3). The $D_2$ and $D_1$ lines at air wavelengths of 766.5 and 769.9 nm (Kramida et al., 2023) are also produced by a similar main excitation mechanism (Swider, 1987; Noll et al., 2019). In Reactions (R3) and (R4), Na only needs to be replaced by K. The reference layer height of 89 km in Table 2 was taken from emission simulations by Noll et al. (2019).

The $D_2$ line is strongly affected by $O_2$ absorption. The reference model transmission is just 0.0026. Satellite-based mea-

surements are not possible as well due to the very strong airglow emission of the (b-X)(0-0) band (Sect. 4.2). Hence, only $D_1$ can be measured. As this line is weak and located in a region crowded by OH and $O_2$ emission, the spectral resolving power of X-shooter is not sufficient and we therefore used 2,299 UVES-based line measurements from Noll et al. (2019) as already mentioned in Sect. 3 and the beginning of Sect. 4. The reference intensity from the resulting climatology is 0.95 R (see Figs. 2 and 3) in agreement with the result of Noll et al. (2019) and also very close to the first published intensity measurement by

Slanger and Osterbrock (2000) of 1.0 R from a HIRES composite spectrum from Mauna Kea. As $D_2$ cannot be measured, we used the theoretical $D_2$-to-$D_1$ ratio of 1.67 from Noll et al. (2019) to obtain a reference intensity of 1.58 R. Hence, the total emission of the K model is 2.53 R.

### 4.5 Atomic oxygen

There are various O lines that contribute to the nightglow emission spectrum. The most prominent line is certainly the green

line at an air wavelength of 557.7 nm (see Fig. 1), which is related to the transition between the two excited states $^1S$ and $^1D$ (see Kramida et al., 2023). The established production mechanism of $^1S$ in the mesopause region (Barth and Hildebrandt, 1961) consists of the atomic oxygen recombination in Reaction (R2) (Sect. 4.2) producing excited $O_2$ and

$$O_2^* + O \rightarrow O_2 + O^*. \tag{R5}$$

The green line emission peaks relatively high compared to the previously discussed emissions (e.g., Yee et al., 1997; von

Savigny and Lednyts'kyy, 2013). As reference height, we use 97 km in agreement with the ESO Sky Model (Noll et al., 2012).



The 557.7 nm line is located in the overlapping region of the X-shooter UVB and VIS arms, which receive light in their wavelength ranges by means of dichroics (Vernet et al., 2011). As the beam splitting shows some variability, there are relatively high flux calibration uncertainties for wavelengths around the green line. On the other hand, it was possible to measure intensity time series in both arms. The resulting climatology-based reference intensities are 156 R for the UVB arm and 170 R for the

VIS arm, which reflects the systematic uncertainties. Assuming that the most reliable value is close to the middle, we averaged the climatologies of both data sets with equal weight, which returned a reference intensity of 163 R (Figs. 2 and 3). This is similar to a value of 153 R that we obtained for the ESO Sky Model with the given solar activity correction for a change from 129 to 100 sfu and a reference intensity of 190 R (Noll et al., 2012).

Other prominent O emissions are the red lines at air wavelengths of 630.0 and 636.4 nm, which are related to the transition

from the lowest excited state $^1$D to the ground state $^3$P (Kramida et al., 2023). These lines are negligible in the mesopause region due to the long radiative lifetime of about 2 min compared to the frequency of relevant collisions. However, they are the most important emission lines in the PALACE wavelength range that originate (at a distinctly lower air density) in the upper thermosphere. The major fraction of the emission is usually emitted between 200 and 300 km and typical peak heights are around 250 km (e.g., Adachi et al., 2010; Haider et al., 2022), but there are significant variations. At nighttime, the main

excitation mechanism (e.g., Link and Cogger, 1988) is triggered by atomic oxygen ions (O$^+$) produced at daytime. Then, the charge transfer reaction

$$O^+ + O_2 \rightarrow O_2^+ + O. \tag{R6}$$

and dissociative recombination with electrons (e$^-$)

$$O_2^+ + e^- \rightarrow O^* + O \tag{R7}$$

can happen, which produces O atoms in the required $^1$D state. In principle, $^1$S populations and 557.7 nm emission can also be produced in this way. However, this ionospheric radiation is usually mucher weaker than the contribution from the mesopause region.

The reference intensities for the 630.0 and 636.4 nm lines derived from the corresponding climatologies are 123 and 41 R, respectively (Figs. 2 and 3). This corresponds to an intensity ratio of 2.99, which is very close to the ratio of the NIST-based $A$

coefficients of theses lines of 3.09 (Kramida et al., 2023). We therefore used these $A$ coefficients to also estimate the reference intensity of the third line of the triplet at 639.2 nm, which can only be generated by an electric quadrupole transition due to the necessary angular momentum change of 2. As a consequence, the resulting intensity is only 0.019 R. The total emission of the red lines amounts to 164 R, which is in good agreement with a value of 161 R for 100 sfu derived from the reference intensity of 190 R and solar activity correction of the ESO Sky Model (Noll et al., 2012). With an average nocturnal intensity similar

to the 557.7 nm line, the combined red lines are relatively strong at Cerro Paranal compared to observing sites at northern mid-latitudes (e.g., Hart, 2019; Mackovjak et al., 2021). Further details are discussed in Sect. 6.2.



Levels with energies higher than $^1$S also contribute to the nightglow emission. They can emit by radiative O recombination, which involves the reaction

$$O^+ + e^- \rightarrow O^* \tag{R8}$$

and subsequent radiative cascades to lower states (Meier, 1991; Slanger et al., 2004). As the emission correlates with the electron density squared, it peaks near the ionisation maximum in the ionospheric F-layer, which is higher than the peak of the red line emission that also depends on the $O_2$ density. We assume a rough difference of 50 km (see Makela et al., 2001), which results in a reference height of 300 km (Table 2). The relevant emission lines can be grouped depending on the spin multiplicity of the upper level (Meier, 1991; Slanger et al., 2004). Quintet states are most important with noteworthy emissions

located at 777, 926, 645, and 616 nm (listed in the order of decreasing intensity) in the PALACE wavelength regime. Each emission feature consists of several components (e.g., three for 777 nm and nine for 926 nm). Examples for emissions related to triplet states are the features located at 845 and 700 nm. Both lines consist of three components.

Using X-shooter spectra, we could measure intensity time series for the features listed above and derive climatologies and reference intensities. However, safe line detections were only possible for favourable observing conditions such as high solar

activity (see Sect. 6.2) in the case of the weaker lines. Moreover, the relatively strong 777 and 926 nm emissions are affected by blending with OH lines. Hence, we corrected the intensities by taking measurements of a nearby unblended OH line with a similar upper level as the blending line and using the OH model from Sect. 4.1 for the estimate of a correction. For example, we subtracted 0.13 times the intensity of OH(9-4)P$_1(N' = 2)$ from the 777 nm intensity.

As the O recombination line measurements were difficult, we built our line model only based on the strongest quintet and

triplet state features, i.e. the emissions at 777 and 845 nm with resulting reference intensities of 16.1 and 6.7 R (see Fig. 3). For the calculation of the strength of the other lines from the NIST database (Kramida et al., 2023) in the PALACE wavelength regime, we used theoretical radiative recombination coefficients from Escalante and Victor (1992). In accordance with Slanger et al. (2004), we selected the data for the optically thin case A and a temperature of 1000 K for this task. NIST lines without coefficients were not considered for our model. The resulting line ratios appear to be realistic as a comparison of the intensities

of the measured features of the same multiplet group indicated. The quintet and triplet groups need to be treated separately as the latter is directly connected to the triplet ground state $^3$P, which causes an increased optical depth. For this reason, the ratio of the 845 and 777 nm is about 2.8 times higher than expected from the recombination coefficients for the optically thin case. However, it is still lower than a factor of 4.8 for the transition to the case with infinite optical depth (Escalante and Victor, 1992), which is in qualitative agreement with results of Slanger et al. (2004) based on line measurements in spectra

from the Echelette Spectrograph and Imager (ESI) and HIRES at Mauna Kea. As the optical depth can depend on the specific line of the triplet group, the model uncertainties are higher than in the case of the quintet transitions. In order to model the strength of the individual multiplet components, we used the NIST $A$ coefficients. Their quality could roughly be checked as some multiplet features were partly resolved. The final O recombination line model comprises 309 lines (Fig. 2) and has an integrated reference intensity of 72 R, where the contribution of the 204 quintet transitions is 55 %.

Finally, the full O line model comprises 313 lines with a summed intensity of 399 R.



### 4.6  Atomic nitrogen

The N transition from the excited $^2$D level to the ground state $^4$S generates a doublet at air wavelengths of 519.8 and 520.0 nm (Kramida et al., 2023). The chemistry of this ionospheric emission is closely related to the chemistry of the red O lines (see Reactions (R6) and (R7)). The $^2$D level population is also produced by dissociative recombination, i.e.

$NO^+ + e^- \rightarrow N^* + O$,                                                 (R9)

where $NO^+$ originates from charge transfer reactions of either $O^+$ with $N_2$ or $O_2^+$ with N (e.g., Khomich et al., 2008). We assume the same reference emission height of 250 km as for the red O lines (Table 2).

The doublet was measured in X-shooter UVB-arm spectra. However, we only considered spectra for widths of the entrance slit not larger than the standard value of 1 arcsec in order to avoid blending of both components. Moreover, the intensity ratio
of both lines showed a dependence on the total intensity. However, Sharpee et al. (2005) found a relatively constant ratio of $1.759 \pm 0.014$ based on ESI and HIRES data from Mauna Kea, which is also consistent with the UVES-based result of 1.760 from the catalogue of Cosby et al. (2006). As only the strongest emissions in the X-shooter-based sample showed a satisfying agreement, there was obviously an issue with the separation of the weak line and relative strong underlying emission (see Fig. 1). Therefore, we corrected the mean ratio of the time series using 1.759 as a reference. The total emission of the doublet remained untouched. From the resulting climatologies, we then obtained reference intensities of 1.91 and 1.08 R for the lines
at 519.8 and 520.0 nm, respectively (Figs. 2 and 3). As a result of this approach, the intensity ratio slightly changed to 1.757. The integrated reference emission of the N line model amounts to 2.99 R.

### 4.7  Atomic hydrogen

Low-latitude nightglow emission is essentially caused by chemiluminescence, i.e. the excited states are produced by chemical
reactions. However, there is also a small contribution from resonant fluorescence to the nocturnal radiation. Although this process requires the absorption of solar photons, such emission is visible if the absorbing atoms are frequent and located higher than the Earth's shadow. These criteria are usually fulfilled for H in the Earth's geocorona that extends thousands of kilometres in altitude. In the PALACE wavelength regime, the lowest Balmer lines are most relevant. The Balmer series is related to the H level with a principal quantum number $n$ of 2. In emission, the first four lines H$\alpha$, H$\beta$, H$\gamma$, and H$\delta$ have
upper levels $n'$ of 3 to 6 and are located at air wavelengths of 656.3, 486.1, 434.0, and 410.2 nm (Kramida et al., 2023). Each transition consists of seven fine structure components. The intensity of the different lines is affected by the H density profile, the altitude of the Earth's shadow, the solar spectrum, radiative transfer by scattering, excitation factors, and radiative cascades in the H atom (Meier, 1995; Gardner et al., 2017). The emission strongly depends on the shadow height due to decreasing H density with increasing altitude. Hence, a fixed reference height as for the chemiluminescent emissions cannot be given.
Moreover, the shadow height depends on zenith distance and azimuth for a fixed time, which requires more information than the PALACE input parameters (Table 1) can provide. As a consequence, we neglected a correction of this effect in the data analysis and the model.





We measured intensity time series for Hα to Hδ in the X-shooter data set. Higher Balmer lines were too faint. As Hβ
to Hδ are weak lines in a wavelength region with complex emission structure due to various $O_2$ bands (Fig. 1), systematic
intensity uncertainties are likely. We therefore checked the mean intensities for very high shadow heights above 80,000 km and
found negative values between $-0.23$ and $-0.30$ R, which we interpreted as systematic errors. In the case of Hα, we obtained
$+1.16$ R, which we did not correct as non-zero emission is expected due to the distinctly stronger multiple scattering for Hα
compared to the other Balmer lines (e.g., Gardner et al., 2017). It is not clear whether Hα emission from the interstellar medium
might contribute. From the calculated climatologies, we derived reference intensities of 5.29, 0.51, 0.19, and 0.11 R for Hα to
Hδ (see Fig. 3), i.e. the total H model intensity, which is dominated by Hα, amounts to 6.10 R. As the shadow height effect
was not considered, these intensities refer to the mixture of lines of sight and observing times of the astronomical observations
in the archive. Very different samples might cause a significant change in the intensity. For the final H model (see Fig. 2),
we had to derive the relative contributions of the seven fine structure components, which cannot be resolved in the X-shooter
spectra. Gardner et al. (2017) reported calculated fractions for Hα. In fact, the two lines related to the transitions from $^2$P to $^2$S
appear to comprise about 95 % of the total emission. Only $^2$P should directly be excited by photons originating from the solar
emission in the far-UV Lyman series line Lyβ that changes $n$ from 1 to 3 in H (Meier, 1995). The upper levels of the other fine
structure components need to be populated by radiative cascades, which is less efficient. We used the intensity fractions from
Gardner et al. (2017) for the other Balmer lines as well. This is a rough extrapolation but the absolute uncertainties are small
for these weak lines.

### 4.8   Helium

Another source of fluorescent emission is He. The most interesting line here is probably at 1,083 nm (e.g., Noto et al., 1998).
It is related to metastable ortho-He (spin multiplicity of 3), which can be produced by $He^+$ recombination. It was not possible
to measure this line in X-shooter spectra due to insufficient resolution to separate it from strong OH emission. The catalogue
of Cosby et al. (2006) includes another ortho-He line at 389 nm with an intensity of 0.54R. However, we did not succeed to
detect this line in a difficult wavelength region with many other emission lines (see Fig. 1). As we could not measure any He
line and derive a climatology, we neglected this minor contribution in the PALACE line model.

## 5   Continuum emission

The line model described in Sect. 4 does not cover airglow emission components without resolved lines in X-shooter spectra.
These pseudo-continua were investigated by Noll et al. (2024a) in all three X-shooter arms, which required the subtraction
of other sky brightness contributions such as scattered moonlight, scattered starlight, and zodiacal light as well as disturbing
thermal emission from the telescope and instrument at the longest wavelengths. This correction was performed by means of
the ESO Sky Model (Noll et al., 2012; Jones et al., 2013). Moreover, the spectra were corrected for atmospheric absorption and
emission in a similar way as discussed in Sects. 2 and 4. The only difference was that the correction was calculated for pixels
instead of lines. The separation of continuum and lines was carried out with the same percentage filters and filter widths as used



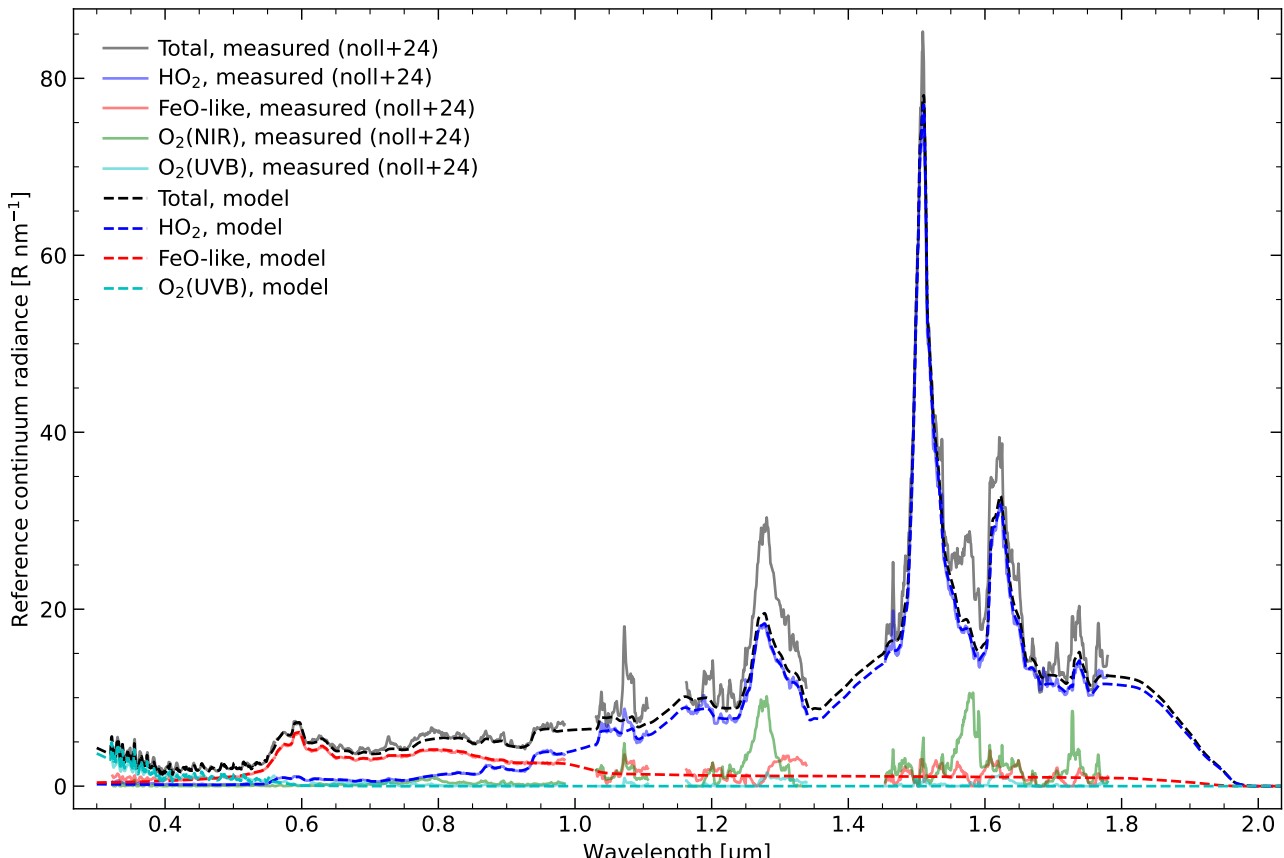

**Figure 7.** PALACE reference continuum model (dashed spectra) compared to X-shooter-based components from Noll et al. (2024a) (solid spectra with gaps). The total (black), HO$_2$ (blue), FeO-like (red), and O$_2$(UVB) emissions (cyan) are shown in both cases. The O$_2$(NIR) residuals (green) are not part of PALACE as they should be covered by the line emission model.

for the line extraction described in Sect. 4. The correction of the van Rhijn effect expressed by Eq. (3) was performed with a reference height of 90 km. For the decomposition of the resulting regridded continuum spectra in the useful wavelength range between 0.3 and 1.8 μm, Noll et al. (2024a) selected a sample of 10,633 high-quality spectra. The mean spectrum is shown in Fig. 7. There are several gaps without data due to strong atmospheric absorption or low-quality marginal ranges of the X-shooter arms. Wavelengths with weak absorption above about 2.0 μm could not be analysed due to too strong telescope-related thermal

emission for a convincing correction. The continuum decomposition was carried out by variability-driven non-negative matrix factorisation applied to the set of spectra. This approach returned an optimum number of four component spectra (see Fig. 7) and corresponding time series of scaling factors that were used for the calculation of climatologies. In the following, we discuss the different continuum components and the modification of the measured spectra for the PALACE continuum model, which is also shown in Fig. 7.



## 5.1 Hydroperoxyl

The predominating component in the X-shooter NIR-arm range is characterised by a conspicuous peak at $1.51\,\mu m$ and weaker
emission features near $1.27$ and $1.62\,\mu m$ as well as a continuum with slowly decreasing flux for shorter wavelengths. This
continuum component was first described and analysed by Noll et al. (2024a), although the residual continuum of the ESO
Sky Model (Noll et al., 2012) roughly reveals its structure. Noll et al. (2024a) found that the emission pattern is very likely the
result of $HO_2$ emission. The strong $1.51\,\mu m$ feature can be identified as the vibrational (200-000) transition of the electronic
ground state $X^2A''$, which changes the $OO-H$ stretching mode (e.g., Becker et al., 1974). The feature near $1.27\,\mu m$ would
mainly be related to a vibronic band with the change of the electronic and vibrational levels from $A^2A'(001)$ to $X^2A''(000)$,
which involves the $O-OH$ stretching mode. The related (000-000) transition, which was bright in laboratory experiments (e.g.,
Becker et al., 1974; Holstein et al., 1983), could not be detected in the X-shooter-based continuum spectrum as the band at
$1.43\,\mu m$ is in a wavelength range with strong water vapour absorption. On the other hand, the $1.62\,\mu m$ feature in Fig. 7 has not
been identified in laboratory studies, so far.

In order to better understand the origin of the emission, Noll et al. (2024a) estimated the total radiation, studied the variability
in the data, and compared the results to simulations of the $HO_2$-related chemistry with an optimised version of the Whole
Atmophere Community Climate Model (WACCM; Gettelman et al., 2019). Moreover, SABER-based retrievals (especially
of H concentrations) were used for the analysis. This investigation revealed that the main production process of $HO_2$ in the
mesosphere (e.g., Makhlouf et al., 1995)

$$H + O_2 + M \rightarrow HO_2^* + M \tag{R10}$$

is also the most important reaction for the nightglow emission. In this way, the spectral distribution of the emission matches
much better than in the case of excitations due to collisions with $O_2$ in the $a^1\Delta_g$ state (Holstein et al., 1983), which can
only produce a minor fraction of the total emission. This constraint is also justified by major differences in the climatological
variability patterns. The WACCM simulations revealed a mean centroid height of the emission related to Reaction (R10) of
$81\,km$, which is in good agreement with rough X-shooter-based estimates derived from the timing of a passing high-amplitude
wave compared to various OH lines (see also Noll et al., 2022). The use of $81\,km$ in PALACE (Table 2) instead of an height
of $90\,km$ that was taken for the data analysis does not cause a systematic error as the van Rhijn factors are almost identical for
the mean zenith angle of the continuum-related sample of $34°$.

The measured $HO_2$ pseudo-continuum in Fig. 7 still shows small-scale structures that could be produced by residuals from
other emissions (especially OH). Moreover, the spectrum only covers a part of the PALACE wavelength range. In order
to obtain a smooth model spectrum, we therefore set suitable data points to remove obvious residuals and to bridge gaps
by means of spline interpolation. As the component spectrum showed significant contamination by $O_2$ emission features at
the shortest wavelengths, we removed any features there and kept the flux almost constant. The $HO_2$ spectrum indicates
significant flux at the longest measured wavelength of $1.78\,\mu m$. As it is not clear how the emission pattern evolves at longer
wavelengths, we slowly reduced the flux up to about $2.0\,\mu m$ and then assumed zero emission. The whole smoothing procedure
is somewhat arbitrary, but the highest systematic uncertainties are related to regions with very strong absorption or strong





emission of different origin, where either the absolute or relative $HO_2$ flux is negligible. Nevertheless, the reference flux
without atmospheric extinction could show significant deviations in wavelength regions without measurements. An example is
the possible emission feature at $1.43\,\mu m$, which is not present in the model. For the output spectrum in the PALACE wavelength
range from 0.3 to $2.5\,\mu m$, we used a wavelength step of $0.02\,nm$, which is consistent with the pixel sizes in the UVB- and VIS-
arm X-shooter spectra ($0.06\,nm$ in the NIR arm). This grid is finer than the $0.5\,nm$ used by Noll et al. (2024a) in order to allow
for a higher resolution in the precalculated reference atmospheric transmission (Sect. 2). In the course of the regridding based
on the spline interpolation, we also changed air to vacuum wavelengths, which are also the standard for the line model. As
last step of the modification of the spectrum, we considered that the mean intensity of the time series and the climatological
reference intensity can slightly differ. From the climatology of the component scaling factors, we obtained a correction factor
of 1.011, which we applied. The final reference spectrum has an integrated flux of $13.1\,kR$. Noll et al. (2024a) reported an
intensity of $11.8\,kR$ for wavelengths between 0.323 and $1.78\,\mu m$ and using linear interpolation for the gaps. For the same
wavelength range, the model spectrum shows a very similar value of $11.7\,kR$.

## 5.2   Iron monoxide and other molecules

The pseudo-continuum in the wavelength range from about 500 to $720\,nm$ with a main emission peak at about $595\,nm$ (Fig. 7)
was already the topic of several studies. Based on spectra from the satellite-based Optical Spectrograph and Infrared Imaging
System (OSIRIS; Evans et al., 2010), ESI at Mauna Kea (Saran et al., 2011), and also X-shooter (Unterguggenberger et al.,
2017) as well as laboratory data (e.g., West and Broida, 1975) and theoretical calculations (Gattinger et al., 2011), the emission
structures could be identified as emission of the FeO 'orange arc' bands, which are related to transitions with the two upper
electronic levels $D\,^5\Delta_i$ and $D'\,^5\Delta_i$ and the ground state $X\,^5\Delta_i$ as the lower level. The excited states should be produced by

$$Fe + O_3 \rightarrow FeO^* + O_2 \qquad\qquad (R11)$$

(Evans et al., 2010).
The decomposition of the X-shooter-based continuum by Noll et al. (2024a) revealed that also emissions at shorter and
longer wavelengths (with a shallow maximum near $800\,nm$) showed a strong correlation with the FeO main feature, which
suggested that additional band systems as investigated by West and Broida (1975) in the laboratory contribute to the FeO-
related component spectrum in Fig. 7. However, WACCM simulations by Noll et al. (2024a) indicated that FeO emission is
probably an order of magnitude too faint to explain the whole component spectrum. Minor additional contributions in the visual
range could be caused by excited NiO (Evans et al., 2011) or reactions of NO and O (Gattinger et al., 2010). Noll et al. (2024a)
also proposed emission from excited OFeOH produced by the reaction of FeOH and $O_3$. This possible nightglow radiation
(with uncertain spectral distribution) could even be stronger than FeO emission. Nevertheless, the integrated emission from
all proposed processes could still be too low. Any other suitable mechanism needs to show similar variations as FeO (see
Sect. 6.2). In order to set the reference height for the continuum component that dominates the X-shooter VIS arm, we focused
on FeO and took the WACCM-simulated mean centroid height of $88\,km$ (Table 2), which is in good agreement with previous
measurements and simulations (Evans et al., 2010; Saran et al., 2011).





In a similar way as discussed for HO$_2$ in Sect. 5.1, we modified the measured continuum with FeO contribution in order to reduce residuals from other emissions and fill gaps. While in the VIS-arm range the changes by the spline interpolation are minor and only worked as a smoothing filter, the UVB-arm range shows an increasing discrepancy with decreasing wavelength in order to remove clear contaminations from O$_2$ band systems. As the measured NIR-arm spectrum indicates large flux variations due to decomposition uncertainties, we strongly smoothed it to obtain slowly declining fluxes up to a maximum wavelength of 2.0 μm. As some peaks could be related to residuals of other emissions, we obtained a 12 % (16 %) lower mean in the range from 1.05 to 1.27 μm (1.45 to 1.78 μm). Although the uncertainties remain high in this regime, non-negligible emissions from FeO or OFeOH appear to be possible. In agreement with Sect. 5.1, we also scaled the resulting model spectrum to be consistent with the climatological mean from the component scaling factors. The correction factor was 1.021. The integrated flux of the resulting continuum component amounts to 2.8 kR. In the reduced range from 0.323 to 1.78 μm, where Noll et al. (2024a) derived 2.9 kR, it would be 2.7 kR. The slight decrease reflects the attempted reduction of residuals from other emissions.

## 5.3 Unresolved molecular oxygen bands

The continuum decomposition of Noll et al. (2024a) also produced two O$_2$-related components (Fig. 7). The component in the X-shooter NIR arm mainly consists of residuals of the (a-X) bands at 1.27 and 1.58 μm (see Sect. 4.2). The other contributions also appear to be caused by line residuals, especially by OH. As these emissions should fully be covered by the PALACE line model (Sect. 4), we did not use this component for the continuum model. This is different for the continuum in the UVB arm, which should mainly be produced by O$_2$ band systems with upper levels of c$^1\Sigma_u^-$, A$'^3\Delta_u$, and A$^3\Sigma_u^+$. As discussed in Sect. 4.2, the model-relevant UVES-based line measurements of Hanuschik (2003) that were interpreted by Cosby et al. (2006) are clearly incomplete. Hence, we had to add a continuum to the model that includes the missing emission from the line measurements. First, we derived a smooth curve from the component spectrum of Noll et al. (2024a) using spline interpolation (see Sect. 5.1). Here, we removed any O$_2$-related band features. Moreover, the flux was assumed to be zero above 680 nm. Then, PALACE was used to calculate the reference line spectrum as shown in Fig. 1 without atmospheric extinction. This model spectrum was added to the smoothed continuum and compared to a representative UVB-arm airglow mean spectrum in the range from 320 to 556 nm with corrected atmospheric extinction and van Rhijn effect. The difference was the basis for the correction of the O$_2$ continuum component. In order to avoid impacts of observational noise, differences in the line-spread function, and wavelength calibration, we applied a moving averaging filter with a width of 2 nm and subsequent automatic spline interpolation with gaps between the selected data points of 1 nm to obtain the final spectrum (Fig. 7) with the wavelength grid of the PALACE continuum model (see Sect. 5.1). The integrated flux is 405 R. Neglecting the extrapolation range between 300 and 320 nm without measured band structures, it would be 339 R. The latter intensity is still significantly larger than a value of 62 R from the line model discussed in Sect. 4.2. The uncertainties in the O$_2$ continuum component are relatively high as the correction of other sky radiance components and atmospheric extinction is difficult in the near-UV.



## 6 Variability

### 6.1 Reference climatologies

The reference line and continuum models described in Sect. 4 and 5 are only valid for the nocturnal annual average, a solar radio flux of 100 sfu, and zenith. In order to calculate spectra for different conditions, a variability model was constructed. As already discussed in Sect. 2, it consists of two-dimensional climatologies of relative intensity $f_0(\texttt{mbin}, \texttt{tbin})$, solar cycle effect $m_{\mathrm{SCE}}(\texttt{mbin}, \texttt{tbin})$, and residual variability $\sigma_{f,0}(\texttt{mbin}, \texttt{tbin})$ for 23 different variability classes, which cover all emission lines and continuum components of the model. Scaling factors for the reference values can then be calculated using Eq. (1), (2), and (3) for the given PALACE input parameters $\texttt{mbin}$, $\texttt{tbin}$, $\texttt{srf}$, and $\texttt{z}$. Table 4 lists the 23 variability classes. The basis for their definition was the involved chemical species and upper electronic, vibrational, and rotational levels.

As most measured climatologies are related to OH, the most detailed variability model could be developed for this molecule. In total, 372 individual climatologies were used to derive 11 reference climatologies. As indicated by Fig. 3, the sample of lines for this purpose is smaller than it was for the derivation of the population model (544 doublets, Sect. 4.1). The quality requirements are higher for full climatologies than for mean values. The detailed investigation of OH line climatologies by Noll et al. (2023b) revealed that differences in the variability mainly depend on the effective emission height and the mixing of cold and hot rotational populations. Emission heights were investigated by Noll et al. (2022) by means of a passing high-amplitude quasi-2-day wave (see also Sect. 6.2) with a vertical wavelength of about 30 km that could be observed in X-shooter spectra and OH emission profiles from SABER. The resulting reference heights are between about 86 and 94 km. They tend to increase with increasing upper vibrational and rotational level, i.e. $v'$ and $N'$, although the dependence on $v'$ gets weaker with increasing $N'$. The contribution of cold and hot populations as visualised in Fig. 5 for the different $v'$ is also important as both populations vary in a different way (Noll et al., 2020, 2023b). As small changes in $T_{\mathrm{cold}}$ can significantly affect the rotational energy where contributions from the cold and hot populations are similar, the transition zone between the energy ranges dominated by the two populations shows increased variability.

In order to cover this pattern, we defined four rotational ranges with $N'$ from 1 to 3 (a), 4 to 6 (b), 7 to 8 (c), and at least 9 (d). Moreover, the height dependence of the emission layers for different $v'$ was also considered. In the ranges a and b, we built classes for pairs of $v'$, whereas the ranges c and d only contain two or one vibrational intervals because of a lack of reliable climatologies from faint high-$N'$ lines (only 6 to 9 in Table 4). However, this classification can also be justified by the decreasing deviations between climatologies of different $v'$ for higher $N'$. In any case, the individual climatologies in each class showed very strong correlation. The reference climatologies were derived from the corresponding set of individual ones by averaging weighted by the square root of the effective intensity. This approach considered that climatologies of brighter lines tend to have a better quality. The square root avoids that only the brightest lines contribute. The intensity can differ by several orders of magnitude as Fig. 3 illustrates.

The quality of individual climatologies depends on the size of the time series. As already discussed in Sect. 4, the climatologies were calculated based on time bins with a length of 30 min and a mininum summed exposure time of 10 min, which significantly decreased the varying quality of exposures of very different length. Moreover, the intervals lowered the discrep-



**Table 4.** PALACE variability classes

| Class | Selection | Sample[a] | Clim.[b] | Lines[c] | Cont.[c] | $1 - r_{\max}^2$ [d] | Nearest[e] | $\langle m_{\mathrm{SCE}} \rangle$[f] | $\langle \sigma_{f,0} \rangle$[g] |
|---|---|---|---|---|---|---|---|---|---|
| OH3a | OH, $v' = [2,3]$, $N' = [1,3]$ | 19,372 | 30 | 189 | 0 | 0.028 | OH5a | +0.116 | 0.297 |
| OH5a | OH, $v' = [4,5]$, $N' = [1,3]$ | 19,461 | 65 | 438 | 0 | 0.010 | OH3b | +0.131 | 0.288 |
| OH7a | OH, $v' = [6,7]$, $N' = [1,3]$ | 19,296 | 62 | 704 | 0 | 0.022 | OH5a | +0.143 | 0.267 |
| OH9a | OH, $v' = [8,9]$, $N' = [1,3]$ | 18,593 | 77 | 947 | 0 | 0.025 | OH4c | +0.160 | 0.256 |
| OH3b | OH, $v' = [2,3]$, $N' = [4,6]$ | 19,492 | 15 | 216 | 0 | 0.010 | OH5a | +0.179 | 0.351 |
| OH5b | OH, $v' = [4,5]$, $N' = [4,6]$ | 19,486 | 35 | 505 | 0 | 0.024 | OH7a | +0.183 | 0.345 |
| OH7b | OH, $v' = [6,7]$, $N' = [4,6]$ | 19,254 | 34 | 792 | 0 | 0.021 | OH9b | +0.189 | 0.318 |
| OH9b | OH, $v' = [8,9]$, $N' = [4,6]$ | 18,320 | 32 | 1,080 | 0 | 0.021 | OH7b | +0.192 | 0.288 |
| OH4c | OH, $v' = [2,5]$, $N' = [7,8]$ | 19,449 | 7 | 480 | 0 | 0.024 | OH7b | +0.191 | 0.286 |
| OH8c | OH, $v' = [6,9]$, $N' = [7,8]$ | 19,118 | 9 | 1,246 | 0 | 0.047 | OH9b | +0.198 | 0.301 |
| OH6d | OH, $v' = [2,9]$, $N' \geq 9$ & OH, $v' = 10$, $N' \geq 1$ | 19,002 | 6 | 15,508 | 0 | 0.117 | OH8c | +0.187 | 0.248 |
| O2a | $O_2$(a-X) bands | 17,479 | 4 | 1,196 | 0 | 0.225 | OH3a | +0.298 | 0.346 |
| O2b | $O_2$(b-X) bands | 15,763 | 7 | 1,303 | 0 | 0.079 | O2Ac | +0.399 | 0.355 |
| O2Ac | $O_2$ A, A′, and c upper states | 7,971 | 1 | 1,590 | 1 | 0.079 | O2b | +0.395 | 0.368 |
| HO2 | $HO_2$ continuum | 17,482 | 1 | 0 | 1 | 0.427 | O2a | +0.042 | 0.346 |
| FeO | FeO (+ other) continuum | 7,971 | 1 | 0 | 1 | 0.143 | Na | +0.086 | 0.343 |
| Na | Na D doublet | 12,935 | 1 | 2 | 0 | 0.143 | FeO | +0.235 | 0.418 |
| K | K D doublet | 2,145 | 1 | 2 | 0 | 0.485 | Na | −0.108 | 0.464 |
| Og | O green line | 17,525 | 1 | 1 | 0 | 0.232 | O2Ac | +0.754 | 0.369 |
| Or | O red lines | 17,488 | 2 | 3 | 0 | 0.054 | N | +1.432 | 0.676 |
| Orc | O recombination lines | 16,450 | 1 | 309 | 0 | 0.522 | OH3a | +2.679 | 0.967 |
| N | N 520 nm doublet | 9,161 | 2 | 2 | 0 | 0.054 | Or | +1.570 | 0.579 |
| H | H Balmer lines | 8,472 | 1 | 28 | 0 | 0.707 | OH4c | +0.172 | 0.346 |

[a] Average size of sample of 30 min bins.

[b] Number of measured climatologies used for the derivation of the class variability.

[c] Number of lines or continuum components with the given class.

[d] Uniqueness or fraction of unexplained variance with respect to the climatologies of relative intensity.

[e] Class with maximum correlation coefficient $r_{\max}$.

[f] Climatology-averaged relative solar cycle effect $m_{\mathrm{SCE}}$ for a change of 100 sfu as given in Eq. (1).

[g] Climatology-averaged relative residual variability $\sigma_{f,0}$ as given in Eq. (2).





ancies in the number of spectra in each X-shooter arm (see Sect. 3). Table 4 shows the resulting average sizes of the time series for each class. They vary between 18,320 and 19,492. The relatively low numbers for the classes `OH9a` and `OH9b` are related

to the fact that a minor fraction of the NIR-arm spectra is only useful up to $2.1\,\mu m$ (Vernet et al., 2011), which affects the (8-6) and (9-7) bands. Nevertheless, all samples were large enough for the derivation of robust climatologies. For this purpose, subsamples of the intensity time series for each `mbin` (month) and `tbin` (local time) combination were selected by including all data points within a circle with a radius of 1 month or 1 hour around the corresponding climatological grid point (see Noll et al., 2023b). Thus, the data sets are overlapping, which smoothes the result. Moreover, a minimum size of the selected subsamples

of 400 was required, which is usually fulfilled. An exception are cells close to twilight or even with daytime contribution. Here, the sample size was optimised by an increase of the selection radius in steps of 0.1 (relative to 1 month or 1 hour) until the threshold size was achieved. As a consequence, the temporal resolution of the climatologies is lower at the margins of the nighttime period. Nevertheless, the effective selection radius for the nocturnal climatologies was only about 1.07 as the cells with the largest radii do not contribute to the average weighted by the nighttime fraction.

The lower limit for the subsample size is important for the calculation of the solar cycle effect, which relies on linear regression using the centred 27-day average of the solar radio flux as reference (Sect. 2). As a regression line is more uncertain than an average, the $m_{\mathrm{SCE}}(\mathtt{mbin},\mathtt{tbin})$ climatologies were more critical for the preselection of suitable OH lines than the $f_0(\mathtt{mbin},\mathtt{tbin})$ ones. Moreover, the $\sigma_{f,0}(\mathtt{mbin},\mathtt{tbin})$ climatologies can easier be disturbed. They were calculated from the standard deviation of the difference between the measured intensities and the intensity model based on Eq. (1). Hence, system-

atic deviations in the data especially due to outliers can cause uncertainties. Therefore, the weighted averaging of individual climatologies further increased the robustness of the measured variability patterns.

The number of available climatologies for all non-OH variability classes was relatively small because of either the difficulty of measurement or the existence of only a single multiplet matching the variability class. The sizes of the data sets used for the calculation of these climatologies were also smaller than in the case of OH (Table 4). The data selection for the few

lines or features tended to be more restrictive. In part, the emissions were relatively faint. In some cases, the slit width of the spectrograph had to be limited to avoid line blending. Overall, this leads to higher uncertainties (and lower temporal resolution) in the reference climatologies, especially with respect to $m_{\mathrm{SCE}}$ and $\sigma_{f,0}$. Exceptions are the few relatively strong emissions (Fig. 3) and classes with strong variations (Sect. 6.2).

For the $O_2$(a-X) bands, only eight X-shooter-based climatologies were used for the population study in Sect. 4.2. For the

derivation of the reference climatology, we further reduced this number to four, only involving the $^{\mathrm{S}}$R and $^{\mathrm{O}}$P lines of the (0-0) band with $N' = 15$ and 17, which showed the best quality. The averaging of these climatologies weighted by reference intensity resulted in an effective $N'$ of 15.7. As a consequence, the reference climatology is only an approximation for lines with low $N'$ that could not be measured. In the case of the $O_2$(b-X) bands, climatologies of P-branch lines of the (0-1) band with $N'$ between 0 and 16 were derived. While $N' = 2$ was already skipped for the population study, $N' = 8$ was also dropped for the

variability analysis due to an outlier-affected $\sigma_{f,0}$ climatology. In the end, seven levels were used for the intensity-weighted averaging of the climatologies, which resulted in an effective $N'$ of 8.0. As variations in the ambient temperature change the rotational level population distribution, the climatologies are expected to deviate for increasing discrepancy in $N'$. This effect is





visible for the comparison of $N' = 0$ and 16 (mainly revealing stronger variability for higher $N'$) but it is relatively weak. The correlation coefficent $r$ for the two intensity climatologies is $+0.975$. Nevertheless, the deviations could be more significant for

very high $N'$ dominated by a hot population (see Fig. 6) as well as higher $v'$, where a maximum of 15 was measured (Slanger et al., 2000). As the most extreme lines are very weak, the systematic uncertainties in the resulting airglow spectrum should be low.

The variability of the $HO_2$, FeO-like, and UVB-arm $O_2$ emissions was investigated by Noll et al. (2024a). Climatologies were derived from the scaling factors of the mean components (Sect. 5). Moreover, there are climatologies based on the

intensities of specific features that are representative of the continuum components. We used the latter as these are more direct measurements with lower systematic uncertainties. The intensities were derived by linear interpolation between two limiting wavelengths in the continuum spectra and subsequent integration of the flux above the line. In detail, we used the results for the main features of $HO_2$, FeO, and $O_2$ in the ranges 1.485 to 1.550 μm, 584 to 607 nm, and 335 to 388 nm, respectively (Fig. 7). The latter range includes several emission features (see Fig. 1). For FeO and $O_2$, the same sample of 7,971 spectra as

for the continuum decomposition (Noll et al., 2024a) was used. As this number is much lower than for OH, this significantly lowered the resolution for the resulting $O_2$ climatology (effective nighttime selection radius of 1.56). In the case of FeO, we decided to set the minimum subsample size from 400 to 200, which increased the noise but kept the climatology at a higher resolution (radius of 1.15). Such changes were not necessary for the relatively strong 1.51 μm feature of $HO_2$, where we could use a relatively large data set of 17,482 bins that was selected by Noll et al. (2024a) optimised for the feature.

Climatologies were measured for both Na lines of the D doublet (Fig. 3). In principle, some variation in the line ratio is expected (Slanger et al., 2005). However, we do not have an unbiased measurement of this variation as we used the line ratio to identify laser contaminations of the $D_2$ line at 589.0 nm (see Sect. 4.3). Hence, we directly took the unaffected $D_1$ climatology as reference for the Na variability class. The sample size of 12,935 bins caused a mild decrease of the resolution (effective selection radius of 1.22). By far, the smallest input sample is related to the weak K $D_1$ line at 769.9 nm. The 2,145 bins were

derived from intensity measurements of Noll et al. (2019) in UVES data (Sect. 3). As a consequence, the resolution of the K-related climatology is relatively low (radius of 1.47), although we only used a minimum subsample size of 100, which increased the noise. Despite these limitations, the climatological patterns for $f_0$, $m_{SCE}$, and $\sigma_{f,0}$ are still meaningful (see Sect. 6.2).

As already discussed in Sect. 4.5, the green O line was measured in the UVB and VIS arm of X-shooter. This resulted in two time series with similar sizes that were converted into climatologies of good quality with very similar patterns. For

the relative intensity, the correlation coefficient $r$ was $+0.999$. The only noteworthy discrepancy was the mean intensity due to issues with the flux calibration (see Sect. 4.5). As reference climatology for the green line, we use the arithmetic mean of both climatologies. The treatment of the two red O lines, which were measured in good quality, was relatively simple. As expected, the resulting climatologies proved to be equal. We therefore obtained the reference climatology for this small variability class from the averaging of the individual climatologies weighted by the intensity. All O recombination lines related

to Reaction (R8) were assigned to the same variability class based on the climatology of the strongest emission feature at 777 nm, which consists of three components. The other measured features (see Sect. 4.5) showed noisy climatologies. In principle, lines related to quintet and triplet upper states could indicate some discrepancies in their variability. According to



Slanger et al. (2004), there might be differences at the lowest intensities, where a fluorescence contribution to the triplet-related states could become significant. The strongest emission feature of the triplet group is at 845 nm. A comparison of the intensity time series did not show a clear effect and the correlation coefficient $r$ for the $f_0$ climatology (which is also robust for the 845 nm emission) turned out to be $+0.998$, which suggests that the reference climatology based on the 777 nm emission is also valid for the other recombination lines.

For the two lines of the N doublet at 520 nm, individual climatologies could be derived (Sect. 4.6). A correlation analysis indicated almost perfect agreement, which confirms the result of Sharpee et al. (2005) that the line ratio should be fixed. We therefore produced a single reference climatology for N by intensity-weighted averaging. As the sample size for the climatology calculation was relatively small (9,161 bins), the temporal resolution is relatively coarse with an effective selection radius of 1.45 for nighttime conditions. As the H$\alpha$ reference intensity contributes 87 % to the total intensity of the H line model (Sect. 4.7) and the summed intensity for the three other Balmer lines is less than 1 R, we used the H$\alpha$ climatology as the reference for all H lines. This approach neglects small systematic changes in the variability patterns for $f_0$, $m_{\mathrm{SCE}}$, and $\sigma_{f,0}$. For $f_0$, the correlation coefficients for H$\alpha$ compared with H$\beta$, H$\gamma$, and H$\delta$ are 0.96, 0.88, and 0.65, respectively. Although the latter value is certainly affected by noise, there is a clear trend. Such discrepancies are expected as the impact of radiative cascades and the opacity of the geocorona for scattering differ, which leads to a different dependence of the intensity on the shadow height (Roesler et al., 2014; Gardner et al., 2017). The sample size for the H$\alpha$ climatology amounts to 8,472 time bins. In order to avoid very low temporal resolution, we calculated the climatology for a minimum subsample size of 300, which resulted in an effective selection radius of 1.34.

## 6.2 Comparison

The resulting reference climatologies of 15 variability classes for the parameters $f_0$, $m_{\mathrm{SCE}}$, and $\sigma_{f,0}$ are shown in Figs. 8, 9, and 10, respectively. All non-OH climatologies are included. Starting in the second row of each figure, they are plotted in the same order as listed in Table 4. In the top row, we focus on three examples for OH, where the variability was discussed by Noll et al. (2023b) in detail. The class OH3a (a) includes the lines with the lowest effective emission heights (Noll et al., 2022). The change in $v'$ without changing $N'$ is indicated by OH9a (b), whereas OH6d (c) is the class for the plain hot population, which comprises the majority of all lines in the PALACE line model (see Table 4). The contours in each climatology are limited to a ground-based minimum solar zenith angle of 100°, which corresponds to complete nighttime up to a height of about 200 km. As already explained in Sect. 2, the given local time (LT) refers to the mean solar time at the longitude of Cerro Paranal.

The climatologies in relative intensity in Fig. 8 show a wide range of variability patterns but also clear similarities. In order to quantify this, Table 4 provides a uniqueness parameter which was derived from the correlation coefficients $r$ for all combinations of $f_0$ climatologies. The maximum $r$ for each class was then used to calculate the parameter by means of $1-r^2_{\max}$. The labels of the best-matching climatologies are also listed in the table. As expected, the OH-related classes show a relatively high similarity with the lowest uniqueness of 0.010 for the pair OH5a and OH3b. Here, the increase of $N'$ for the latter appears to be almost completely compensated by the lower $v'$. The lines of both classes should have very similar emission heights (see Noll et al., 2022).





**Figure 8.** Climatologies of intensity relative to the annual mean for a solar radio flux of $100\,\mathrm{sfu}$, $f_0$, with respect to local time (mean solar time at Cerro Paranal) and day of year for 15 PALACE variability classes as defined in Table 4. The coloured contours only show data with a minimum solar zenith angle of $100°$. The lighter colours at the left and right margins mark the repeated patterns of December and January.







**Figure 9.** Climatologies of the relative intensity change by an increase of the solar radio flux by $100\,\text{sfu}$ or $1\,\text{MJy}$, $m_{\text{SCE}}$, with respect to local time and day of year for 15 PALACE variability classes as defined in Table 4. The solar cycle effect is provided relative to the corresponding relative intensities $f_0$ in Fig. 8.





**Figure 10.** Climatologies of the relative residual variability (after subtraction of the climatological intensity variations considering the solar cycle effect), $\sigma_{f,0}$, with respect to local time and day of year for 15 PALACE variability classes as defined in Table 4. The standard deviation is provided relative to the corresponding relative intensities $f_0$ in Fig. 8.





We start the discussion of the variability patterns in Fig. 8 with emissions originating in the mesopause region, which cover the plots from (a) to (k). Here, $HO_2$ in (g) indicates a high uniqueness of 0.427. As already described by Noll et al. (2024a), the climatology shows high $f_0$ at the beginning of the night and seasonal maxima in January and July/August. This pattern

correlates well with SABER H density retrievals (Mlynczak et al., 2014), which confirms Reaction (R10) as crucial for the production of excited $HO_2$. In the reaction, only H is a volatile minor species. It is mainly produced by $H_2O$ photolysis at daytime and consumed at nighttime by different processes, which would explain the nocturnal trend. There is no other known nightglow process in the mesopause region that predominantly depends on H. In Fig. 8h, the climatology derived from the main FeO emission feature (Sect. 6.1) is very different. As also analysed by Noll et al. (2024a), it is characterised by an

almost opposite seasonal variation with maxima in April/May and October and the lack of a clear nocturnal trend. The main driver for this variability is ozone ($O_3$) in Reaction (R11). Similar variations were found in SABER-based $O_3$ densities for the region around Cerro Paranal (Noll et al., 2019). The impact of Fe atoms on the climatology is the higher intensity in austral autumn/winter compared to the other seasons (Feng et al., 2013; Unterguggenberger et al., 2017). The source of metals in the mesopause region is their ablation from heated meteoric dust, which produces permanent metal layers (e.g., Plane et al., 2015).

As this also includes Na atoms and the production of excited $^2P$ states also involves $O_3$ in Reaction (R3), the similarity of the $f_0$ climatologies in (h) and (i) is expected (Unterguggenberger et al., 2017). However, the emission of the other alkali metal K in (j) shows clear discrepancies, especially a remarkable maximum in June at the end of the night that was first observed by Noll et al. (2019). The climatology has a very high uniqueness of 0.485 based on the relation to Na, although the reactions leading to the nightglow emission are similar. However, the whole reaction networks of both species show differences (Plane

et al., 2014) that could explain the discrepancies.

The production of excited OH requires H and $O_3$ according to Reaction (R1). As a consequence, the seasonal variability patterns of both species cancel out in the plots (a) to (c) for most local times. An exception is the beginning of the night, where a semiannual variability pattern with maxima near the equinoxes as expected for $O_3$ (see the plots for FeO and Na) is present, although relatively weak for the hot population lines in (c). With decreasing effective emission height (Noll et al.,

2022), this structure becomes stronger. This trend and the LT-related pattern can be explained by the strong decrease of the mean O concentrations in the lower parts of the OH emission layer (e.g., Smith et al., 2010). As O radicals are mostly produced by photolysis of $O_2$ at daytime, this causes only short lifetimes of O below about $83\,\mathrm{km}$ (Marsh et al., 2006; Noll et al., 2023b) due to its nocturnal consumption by the production of ozone via

$$O + O_2 + M \rightarrow O_3 + M \tag{R12}$$

and the subsequent creation of excited OH. In order to explain the observed seasonal pattern, the variability of the $O_3$ formed near $80\,\mathrm{km}$ needs to be stronger than in the case of the H density (see $HO_2$ in (g) as a tracer), which peaks at these heights (Mlynczak et al., 2014; Noll et al., 2024a). As described by Noll et al. (2023b), all OH climatologies show a local maximum in May close to dawn and a global minimum in the middle of the night in August/September. These structures are related to the influence of solar thermal tides with periods of integer fractions of $24\,\mathrm{h}$ (Forbes, 1995; Smith, 2012) on the density, temperature,

and O concentration. The effect of tide-induced vertical transport on O is amplified by the strong vertical density gradient.



According to Noll et al. (2023b), the tidal modes (and/or the interacting perturbations) relevant for the most conspicuous variability structures should have a relatively long vertical wavelength as there is no significant change in the features for different effective emission heights of the OH lines.

The tidal features as visible in (c) can also be observed in the $O_2$-related climatologies in (e) and (f), which also depend on a reaction involving O (R2). The differences at the beginning of the night are probably related to the several kilometres higher emissions (causing a lower impact of heights with relatively short lifetime of the O radicals). The $f_0$ climatologies of the O2b and O2Ac classes show a good agreement with a uniqueness of 0.079 in Table 4. The climatology of the green O line in (k) also indicate similarities (0.232 with respect to O2Ac). This is not unexpected as the chemistry of these emissions is closely related (see Sect. 4.5). On the other hand, the climatology for the $O_2$(a-X) bands in (d) shows clear discrepancies. While the tidal features in the second half of the night are still weakly visible, the beginning of the night reveals a strong exponential intensity decrease. As already discussed in Sect. 4.5, this component is caused by the slow decay of an $a^1\Delta_g$ population produced by $O_3$ photolysis at daytime (Noll et al., 2016).

Figure 8 shows that the climatologies of the ionospheric red O lines (l) and the N doublet at 520 nm (n) are very similar (uniqueness of 0.054), which demonstrates the close relation of the chemistry of both emissions (Sects. 4.5 and 4.6). The highest intensities are found between April and September after dusk. If the minor ionospheric contribution of the green O line (k) shows a similar pattern, it might explain differences in comparison to the $O_2$ emissions in (e) and (f) at the beginning of the night. The intensities of the ionospheric emissions clearly decrease in the course of the night with the exception of two local maxima in March and October at about 02:30 LT. The general trend is expected as photoionisation is limited to daytime, which causes a decrease of the ion density due to recombination at nighttime. However, ion dynamics can also lead to intensity increases. Cerro Paranal is about 15° south of the magnetic equator, where a fountain effect exists that lifts ions to high altitudes, where recombination is slow (e.g., Immel et al., 2006). At nighttime, the propagation reverses depending on the magnetic field lines and the season-related direction of the meridional neutral wind. Then, the airglow emission rate depends on when this ion reservoir reaches the bottom side of the F-layer, which can lead to effects such as the midnight brightness wave (e.g., Adachi et al., 2010; Haider et al., 2022). In any case, this equatorial ionisation anomaly causes unusually high ionospheric line intensities at Cerro Paranal. Interestingly, the $f_0$ climatology of the O recombination lines as derived from the 777 nm emission (m) shows strong discrepancies. It is the pattern with the strongest variations of all classes. A reason is the dependence on the square of the electron/ion density (e.g., Makela et al., 2001). The pattern only indicates high intensities at the beginning of the night in March and October, which brackets the high intensity range of the red O and N lines. Moreover, the 02:30 LT maxima of the latter appear to be located at the end of the nocturnal decrease of the 777 nm emission. These structures seem to indicate the descending ion propagation as the layer of the O recombination lines is about 50 km higher (Table 2).

The most unique $f_0$ climatology is related to H (0.707 in Table 4). This is not unexpected as the radiation mechanism is fluorescence instead of chemiluminescence (Sect. 4.7). This causes a strong dependence of the emission on the shadow height as illustrated by Fig. 8o, which shows high intensities after dusk and before dawn. In addition, there also appears to be variations in the H column density. An example is the local maximum in June at midnight.



As discussed in Sect. 6.1, the derived $m_{SCE}$ climatologies have higher uncertainties due to their dependence on a regression analysis instead of a robust average. Hence, the climatologies in Fig. 9 tend to look noisier than those in Fig. 8. Nevertheless, the existence of relatively strong features in the plots (but not their exact shape and strength) is relatively safe. For OH, the average regression uncertainty is $0.05\,\mathrm{MJy^{-1}}$ (or per $100\,\mathrm{sfu}$), whereas this value doubles for metal-related emissions with

rather small time series (see Table 4). Ionospheric emissions show the highest absolute uncertainties (up to $0.18\,\mathrm{MJy^{-1}}$) but the lowest uncertainties relative to the corresponding $m_{SCE}$. The resulting error in the PALACE output intensity is distinctly lower than 10% in most cases since the deviation of the solar radio flux (SRF) from the reference of $100\,\mathrm{sfu}$ is usually much lower than $100\,\mathrm{sfu}$. For the whole X-shooter data set that almost covers an entire solar cycle, the 27-day averages only ranged from 67 to $166\,\mathrm{sfu}$.

The solar cycle response is far from homogeneous in Fig. 9. As discussed by Noll et al. (2023b), the OH $m_{SCE}$ climatologies show a conspicuous maximum in the second half of the night around July. Moreover, its location with respect to local time depends on the effective emission height. Earlier LTs correspond to larger altitudes, which can be explained by the upward propagation of oblique wave fronts. As could be shown by Noll et al. (2023b) also considering other OH airglow properties, the LT shifts are consistent with a period of $24\,\mathrm{h}$ and a vertical wavelength of about $30\,\mathrm{km}$. These properties describe the migrating

diurnal tide (e.g., Forbes, 1995), which is the most important tidal mode at low latitudes (e.g., Smith, 2012). The diurnal tide obviously amplifies the solar cycle effect (SCE) depending on its phase. Tide-induced downward transport of O radicals with enhanced production during high solar activity to the OH-relevant heights would be an explanation.

An increased SCE in the middle of the year is also visible for the other emissions in the mesopause region. Although the exact location of the centroid of the SCE is difficult to determine, the $O_2$-related emissions in (d) to (f) and also the green O

line in (k) appear to be consistent with the OH-related scenario as these emissions seem to show LT centroids at earlier times as expected for higher effective emission heights (Table 2). The green line as highest emission would need to peak in the evening, which is the case. For the metal-related emissions in (h) to (j) and $HO_2$ in (g), the trend in LT is less clear, which might be related to the less important role of O radicals. Concerning changes in the seasonal patterns, the latter emissions tend to partly show negative SCEs in the first few months of the year. On the other hand, the $O_2$-related emissions in (d) to (f) appear to

indicate additional maxima in March and December. This trend can already be seen in the OH hot population climatology in (c) and a maximum in March is also visible for the green O line in (k). As the LTs of these features do not significantly differ, they do not appear to be linked to the diurnal tide.

Apart from the climatological pattern, the effective SCE is interesting for a comparison. Hence, Table 4 shows $\langle m_{SCE} \rangle$, which is the SCE averaged for the nighttime contour area in the plots. Note that the value can slightly differ if the effect

is calculated with respect to the reference intensity of the full climatology instead of the intensities for each grid point. For OH, $\langle m_{SCE} \rangle$ increases with increasing emission height (Noll et al., 2022), at least for low $v'$ and $N'$, from about $+0.12$ to $+0.20\,\mathrm{MJy^{-1}}$. According to Noll et al. (2023b), this trend can partly be explained by the position of the maximum in July, which tends to be not fully covered by the night for those emissions with the lowest SCEs as in Fig. 9a. The OH-based results are in good agreement with earlier ones related to UVES data from 2000 to 2015 (Noll et al., 2017). On the other hand, the

analysis of Noll et al. (2012) based on FORS 1 data from 1999 to 2005 with a high average SRF of $150\,\mathrm{sfu}$ did not show



a significant effect. The result of $+0.30\,\mathrm{MJy}^{-1}$ for O2a is consistent with the SABER-based latitude-dependent study by Gao et al. (2016) for the $O_2$(a-X)(0-0) band. The same investigation also revealed good agreement in terms of OH. The O2b and O2Ac classes show even higher SCEs of about $+0.40\,\mathrm{MJy}^{-1}$. However, this value is still lower than $+0.63\,\mathrm{MJy}^{-1}$ as returned by Noll et al. (2012) for $O_2$(b-X)(0-1). The relatively strong effects are certainly related to the fact that Reaction (R2)

involves two O radicals. In the case of the $557.7\,\mathrm{nm}$ line, there are even three O radicals required because of the combination of Reactions (R2) and (R5). Hence, a resulting $\langle m_{\mathrm{SCE}} \rangle$ of $+0.75\,\mathrm{MJy}^{-1}$ is not unexpected. The ESO Sky Model (Noll et al., 2012) uses $+0.87\,\mathrm{MJy}^{-1}$. As reported by Zhu et al. (2015), different satellite- and ground-based studies returned typical values of $+0.4$ to $+0.6\,\mathrm{MJy}^{-1}$ for low latitudes (but not covering Cerro Paranal). The same study also showed an increasing SCE with increasing altitude for the green O line.

As discussed by Noll et al. (2024a), $HO_2$ is very different with an SCE close to zero. The low emission altitudes and the independence of O may explain this result, which was also confirmed by WACCM simulations. The same study also revealed the weakly positive effect for FeO ($+0.09\,\mathrm{MJy}^{-1}$) in Table 4. Other metal-related emissions also have rather low effects. The emission of Na D returned a moderate value of $+0.24\,\mathrm{MJy}^{-1}$, whereas Noll et al. (2012) measured $+0.11\,\mathrm{MJy}^{-1}$. As already derived by Noll et al. (2019), the mean effect for K $769.9\,\mathrm{nm}$ emission is even negative with $-0.11\,\mathrm{MJy}^{-1}$. WACCM

simulations of the Na, Fe, and K metal layers by Dawkins et al. (2016) indicated a similar order of the SCEs but more negative in all cases. Hence, the metal concentrations appear to be an important driver for the low $\langle m_{\mathrm{SCE}} \rangle$ values.

Very strong positive solar impacts are present for the lines originating in the ionosphere, where photoionisation depends on extreme UV photons from the Sun. The red O and N lines show about $+1.5\,\mathrm{MJy}^{-1}$, which is much higher than $+0.68\,\mathrm{MJy}^{-1}$ for the red O lines in the ESO Sky Model. The discrepancies are most likely related to the different sample properties. The

sample of Noll et al. (2012) covered parts of an earlier solar cycle with relatively high monthly SRF between 95 and $228\,\mathrm{sfu}$. There might be a flattening of the SCE for very high SRF not present in the X-shooter data set (maximum of $166\,\mathrm{sfu}$). Moreover, the relatively small number of 1,186 FORS 1 spectra could have caused sampling issues. However, the $m_{\mathrm{SCE}}$ climatology in Fig. 9l indicates SCEs higher than $+1\,\mathrm{MJy}^{-1}$ for most of the year, although there are remarkable negative effects at the beginning of the night in austral summer. As the strongest positive effects are located around midnight, this leads to a significant

change of the intensity pattern in Fig. 8l for high SRF. Then, the midnight brightness wave appears to be more powerful. The dependence of the O recombination lines on the squared $O^+$ (or electron) density, leads to the most extreme $\langle m_{\mathrm{SCE}} \rangle$ of $+2.7\,\mathrm{MJy}^{-1}$. The pattern in Fig. 9m shows some similarities with the other ionospheric lines. However, the SCE is always positive with a minimum where also the mean intensity indicates a minimum. The maximum SCE after dusk in May is located in a region in Fig. 8m with a strong intensity gradient, i.e. small changes in the seasonal distribution can have large effects.

In contrast to the ionospheric lines, only a moderately positive effect is present for $H\alpha$ emission ($+0.17\,\mathrm{MJy}^{-1}$). This is about 3 times lower than found by Nossal et al. (2012). However, their data were from Wisconsin in the United States and only covered 22 nights between 1997 and 2008. The $m_{\mathrm{SCE}}$ climatology in Fig. 9o indicates the largest solar impact in the second half of the night.

The residual variation $\sigma_{f,0}$ as derived after the subtraction of the SCE-corrected reference climatological model is shown in

Fig. 10 for the different variability classes. As $\sigma_{f,0}$ corresponds to the standard deviation, which enhances the impact of outliers,





the uncertainties are higher than in the case of the $f_0$ climatologies. Measurement errors could cause significant deviations. Nevertheless, the main structures should be reliable in most cases.

Noll et al. (2023b) analysed the OH-related $\sigma_{f,0}$ climatologies in detail. As shown in (a) to (c), the climatologies reveal a remarkable seasonal pattern with maxima in January and June/July. In a similar way as discussed for the SCE, the centroids of

the variability in the maximum months depend on LT. The migrating diurnal tide also affects the perturbations causing these variations. For this reason, the features in (c) occur earlier than in (a). A separation of 4 h would correspond to 5 km in altitude. In order to better understand the spectrum of periods that causes these structures, Noll et al. (2023b) studied the residual variability as a function of time difference. This approach revealed that periods shorter than 1 day dominate in the middle of the year in austral winter, which suggests that the emission variations are either directly related to (especially long-period)

gravity waves (GWs, e.g., Fritts and Alexander, 2003) and/or the tidal modes are modified by interaction with GWs. A survey of GW-related literature with a focus on South America (e.g., Alexander et al., 2015; Ern et al., 2018; Cao and Liu, 2022) indicated that the main origin of the waves is at higher latitudes along the Andes, which is a global GW hot spot in austral winter. The waves then reached Cerro Paranal either directly or indirectly via secondary waves. The $\sigma_{f,0}$ maximum in January also indicates GW contributions, but especially from deep convection across the Andes. However, the main reason for the

austral summer maximum is the occurrence of quasi-two-day waves (Q2DWs), the strongest waves at low southern latitudes (e.g., Ern et al., 2013; Gu et al., 2019; Tunbridge et al., 2011). They are usually active only a few weeks in austral summer. Noll et al. (2022) used the strong event in 2017 to study OH effective emission heights (see Sect. 6.1). The same approach was applied by Noll et al. (2024a) for the emissions of $HO_2$ and FeO.

We therefore expect to see GW and Q2DW features in the $\sigma_{f,0}$ climatologies for different emissions in the mesopause

region. Indeed, the seasonal structures show similarities in Fig. 10. Good examples are $O_2$(b-X) (e) and $HO_2$ (g). In contrast, K (j) shows very different features, which might be related to the very small sample size for this weak emission. In the case of $O_2$(a-X) (d), the situation in the first half of the night is probably complicated by the contribution of emissions from relatively low heights related to $O_3$ photolysis (see Sect. 4.2). In the same way as Noll et al. (2024a), we checked the period-dependent contributions to winter and summer, which confirmed the OH-related findings. In principle, the centroid LT should match the

height of the corresponding emission layer in all seasons. Although this is not well constrained in many cases, the trends are promising. $HO_2$ peaks later than, e.g., FeO and $O_2$(b-X). In this context, the result for the green O line (k) is most difficult to interpret as the winter and summer maxima are at very different LTs. As the quality of the data of this line is relatively good and the reference height is close to the top of the mesopause region, these discrepancies might point to height-dependent changes in the importance of GWs and Q2DWs.

In Table 4, we also list the night averages of $\sigma_{f,0}$ for each class. Here, the OH hot population class `OH6d` indicates the lowest relative residual variability of 0.25. On the other hand, the intermediate-$N'$ class `OH3b` shows the maximum for OH of 0.35. This result reflects the variable mixing of cold and hot rotational populations, which causes additional variability for lines in classes like `OH3b` (Noll et al., 2023b). As discussed in Sect. 6.1, this effect had an influence on the classification scheme for OH. The other emissions of the mesopause region tend to have similar or somewhat higher $\langle\sigma_{f,0}\rangle$ as/than `OH3b`. The maximum

of 0.46 for K might be partly caused by noise, although the other alkali metal Na indicates the second highest value of 0.42.





The results for OH, $O_2$(b-X), Na, and green O line are roughly consistent with those of Noll et al. (2012) based on FORS 1 data.

The ionospheric lines indicate the largest $\langle\sigma_{f,0}\rangle$ with a maximum of 0.97 for the O recombination lines, which is comparable to the results for the SCE. For the red O lines, Noll et al. (2012) obtained values of about 0.9, which is higher than the X-shooter-based mean standard deviation of about 0.68. As indicated by a comparison of Figs. 8 and 10, the high variability mainly originates from combinations of month and LT with low average intensity. The maximum in $\sigma_{f,0}$ at midnight in December for the red O (l) and N (n) lines corresponds to a minimum in the corresponding $f_0$ climatologies. Typical reasons for variability in ionospheric airglow emissions are plasma instabilities that cause effects like plasma bubbles (e.g., Makela, 2006) and wave-like disturbances with and without links to lower atmospheric layers (e.g., Paulino et al., 2016).

Fluorescent H emission only shows moderate residual variability (0.35). The highest $\sigma_{f,0}$ are located near the minima in $f_0$ (Figs. 8o and 10o). Hence, the H density could be more variable at higher altitudes. However, the larger variation in the shadow height due to changes in the line of sight (which was not corrected, see Sect. 4.7) could also play a role.

## 7 Evaluation

The model development described in Sects. 4 to 6 was complex. Although many checks were performed for the different steps of the procedure, it is not clear how well the final PALACE model represents the observed data. Therefore, we carried out an evaluation of the model based on high-quality X-shooter spectra with very low noise level and without obvious systematic errors in the central parts of the different arms. Noll et al. (2024a) selected such a sample (consisting of 10,633 spectra) for the investigation of the nightglow continuum. We further reduced this by excluding spectra where the exposures were split depending on the X-shooter arm. In this way, we avoided the averaging of spectra before the analysis. The resulting sample contained 7,195 spectra in all arms. We also neglected spectra taken before 5 February 2010 as they showed systematic effects in certain near-UV wavelength ranges connected with the different echelle orders. These calibration issues would have increased the scatter with respect to PALACE. As a consequence, the final sample comprised 6,874 spectra. As the night-sky radiation also involves scattered moonlight, scattered starlight, and zodiacal light and the thermal radiation of telescope and instrument also disturbs, non-airglow contributions had to be subtracted. Here, we used the corresponding spectra of the ESO Sky Model (Noll et al., 2012; Jones et al., 2013) that were already calculated by Noll et al. (2024a) for the continuum study. For the resulting X-shooter airglow spectra (divided into three arm-dependent parts), we then calculated PALACE model spectra in air wavelengths under the consideration of the observing conditions described by the input parameters z, mbin, tbin, srf, and pwv (Table 1).

For the evaluation, we binned the PALACE and X-shooter spectra by using wavelength-dependent bins with a size of about 0.001 times the wavelength. This step is important as the modelled simple Gaussian line profiles depending on the parameter resol cannot reproduce the real line-spread function, which can cause significant residuals. This is even an issue for more sophisticated kernels due to the pixelation of the data and wavelength calibration uncertainties. Our choice of the bin size corresponds to a resolving power of 1,000, which is sufficiently low compared to values of 3,200 to 18,400 for the X-shooter







**Figure 11.** Evaluation of PALACE (solid lines) and the airglow component of the ESO Sky Model (Noll et al., 2012) (dash-dotted lines) based on a comparison to 6,874 X-shooter spectra after the subtraction of non-airglow components calculated by means of the ESO Sky Model. Results are provided for the wavelength range from 0.32 to 1.86 µm divided into three ranges related to the different X-shooter arms. The data points correspond to mean values in bins with a width of about 0.001 times the central wavelength. The green and light green curves show the mean deviation of the two models calculated for the different observing conditions from the X-shooter airglow emission relative to the average of the latter. The black and grey curves display the ratio of the mean residual variability of the models and the standard deviation of the difference between model and measured data. The dashed magenta curve indicates the average fraction of the PALACE airglow emission compared to the full sky brightness that also considers the Sky Model non-airglow components.





data (Vernet et al., 2011). The difference of the averaged fluxes of the PALACE and X-shooter spectra in each wavelength bin was then analysed by the calculation of mean and standard deviation for the whole sample. The results were refined by the application of a $\sigma$-clipping approach for the flux data of each bin. This was important as systematic outliers in the observed data are possible. In particular, strong lines of astrophysical origin (that were not an issue for the continuum study of Noll et al. (2024a)) can have a significant effect. However, the wavelength-dependent fraction of rejections was always lower than 3%. In most cases, only a few spectra were excluded for a given bin. Nevertheless, the sample was about 22% smaller for wavelengths beyond 2.1 µm due to a special set-up for background reduction in the instrument at shorter wavelengths that blocked this range (Vernet et al., 2011).

The results of the analysis are shown in Fig. 11 for the wavelength range from 0.32 to 1.86 µm. Longer wavelengths than about 1.8 µm are difficult to study because of increasing thermal emissions from the instrument and the atmosphere. The situation further deteriorates due to the presence of relatively strong absorption bands, which affect the airglow, whereas the emission of the instrument is not significantly absorbed. First, we discuss the relative deviation, which is the mean difference between PALACE and X-shooter fluxes divided by the mean X-shooter flux. As desired, the green solid curve is close to 0, although with some scatter. For the wavelength ranges from 0.32 to 0.56 µm (UVB arm), 0.56 to 1.02 µm (VIS arm), and 1.02 to 1.78 µm (NIR arm), we obtained mean relative deviations of only $+0.053$, $-0.008$, and $+0.032$, respectively. The latter value is not significantly affected by the few outliers near 1.4 µm, which are related to strong water vapour absorption and hence very low fluxes. The impact of the instrument-related systematic discrepancies at wavelengths above 0.99 µm in the VIS arm (overlap with NIR arm) can also be neglected ($+0.002$ without this range).

The relative scatter is also shown in the figure (black solid curve). This quantity represents the ratio of the sample-averaged residual variability from the model (Sect. 2) and the mean standard deviation from the difference of the PALACE and X-shooter fluxes. The values should ideally be close to 1 for our high-quality X-shooter test sample. For the three wavelength ranges defined above, we calculated mean ratios of 0.769, 0.983, and 1.002, respectively. The former value for the UVB arm is distinctly below 1. Figure 11 shows particularly low values down to about 0.5 in the range between 420 and 520 nm. The main reason for these discrepancies is the fact that other sky radiance components had to be subtracted from the X-shooter spectra. As these components also have uncertainties, the modelled airglow variability is lower than the measured variability. This effect is particularly strong where the fraction of airglow emission compared to the total emission is relatively low. For illustration, this ratio is also plotted in Fig. 11 (magenta dashed curve). Between 420 and 520 nm, its mean value is just 0.32 (with a minimum of 0.25), i.e. airglow is a minor contribution. If we only select the 34% of bins with an airglow fraction of at least 0.5, the mean relative scatter already increases to 0.961, which is more convincing. The VIS-arm range is also partly affected. There, a focus on a minimum airglow fraction of 0.5 causes a slight increase of the mean relative scatter to 1.024. While scattered moonlight and zodiacal light can have strong impacts in the visual range, airglow is the dominating nocturnal radiation source in the NIR-arm range up to about 1.78 µm as demonstrated by the magenta dashed curve in (c). For longer wavelengths (where no nightglow continuum could be measured), the model accuracy remains uncertain, although the only relevant emission lines are related to the well-modelled OH radical (Fig. 1).



In the ranges where an evaluation was possible, the performance of PALACE is convincing as the typical deviations for the two investigated properties are only of the order of a few per cent. In principle, a relative scatter slightly above 1 could even be expected as the model standard deviations for all lines and continua were just averaged for simplicity (see Sect. 2). In particular, wavelength regions with similar contributions from variability classes with very different climatologies should be affected. An example would be ranges in the VIS arm where the OH and FeO-like classes (see Sect. 6) are important. Indeed, the potentially affected regions indicate a slightly increased relative scatter in Fig. 11b. Concerning the relative deviation, the UVB-arm regime in (a) shows a small positive offset, which tends to increase with decreasing wavelength (+0.075 between 320 and 360 nm). This result seems to reflect the difficulty to measure the nightglow in a range where other sky brightness components are relatively bright and atmospheric scattering is relatively strong but uncertain due to variable aerosol properties. There might be additional bias in the model which is not visible in Fig. 11 as PALACE is based on the same X-shooter data that were used for the evaluation. However, a complex flux calibration approach was applied (Noll et al., 2022) that should limit the mean uncertainties to a few per cent. Moreover, the comparison of OH populations from X-shooter data and the UVES mean spectrum from Noll et al. (2020) based on data from 2000 to 2015 (Noll et al., 2017) showed good agreement (see Sect. 4.1). The deviations were only of the order of a few per cent and mainly related to real population differences as indicated by their dependence on the upper vibrational level. Finally, it is possbile that significant long-term changes in the airglow mean intensity and variability occur. For recognising such trends, the covered period of the X-shooter data of 10 years is too short.

PALACE was developed for the same location as the ESO Sky Model, which is briefly described in Sect. 1. Hence, it is interesting to compare both models. In the previous sections, we have already made comparisons with respect to the reference intensities and climatologies of the green and red O lines, the Na D doublet, OH, and $O_2$(b-X)(0-1) that were measured by Noll et al. (2012) in low-resolution FORS 1 data. There are clear similarities, although some discrepancies are also observed (see Sects. 4 and 6.2). Some uncertainties are related to the very different solar radio flux ranges for the X-shooter and FORS 1 spectra. Moreover, the ESO Sky Model climatologies are too coarse (18 compared to 134 useful grid points) for a detailed comparison. An evaluation of the mean nightglow continuum of the ESO Sky Model (derived from the FORS 1 data and a few early X-shooter NIR-arm spectra) was already performed by Noll et al. (2024a), which indicated good agreement up to about 800 nm and major disrepancies at longer wavelengths, although the $HO_2$ 1.51 μm peak (Fig. 7) was visible (see also Jones et al., 2019). Noll et al. (2012) measured the continuum variability only at 543 nm, which would best fit to the FeO class. However, an SCE of $+0.61 \, \text{MJy}^{-1}$ and different climatological patterns as for Na do not match the expectations (see Table 4). There might also be significant contributions from $O_2$ and other emissions. The low resolving power of the FORS 1 spectra and the lack of near-IR data also affected the line model, which is only based on simple theoretical calculations beyond 925 nm.

In order to better understand the overall performance of the widely used ESO Sky Model, we carried out the same comparison to the selected X-shooter data set as for PALACE. As the removal of the non-airglow components in the X-shooter spectra already required the calculation of the Sky Model, we could directly use these spectra. Their line-spread function is a combination of boxcar and Gaussian optimised for the different arms and slit widths. However, this difference in comparison to PALACE (only Gaussian) should not matter due to the binning of the wavelength grid. The results of the analysis are also presented in Fig. 11 (dash-dotted curves).




For the UVB-arm range, relative deviation (light green) and scatter (grey) are $+0.239$ and $0.842$ on average. If only bins with an airglow fraction of at least 0.5 are considered, these values change to $+0.067$ and $0.989$. Thus, the relative scatter appears to be realistic and similar to PALACE, whereas the relative deviation is clearly too high. As illustrated by Fig. 11a, the largest positive deviations correlate with low airglow fractions. Hence, there could have been issues with the separation of airglow and other radiance components in the FORS 1 spectra. Systematic uncertainties in the flux calibration of the data that were collected by Patat (2008) could contribute here.

In the VIS-arm range, the average relative deviation amounts to $+0.198$, which is mostly driven by the wavelength range between 800 and 930 nm, where even deviations up to about 2.0 are visible (b). As wavelengths with strong line emission (see Fig. 1) show values near 0, this is caused by the unrealistic FORS 1-based airglow continuum in this range (Noll et al., 2024a), where the spectral resolution and quality of the flux calibration were obviously insufficient. The average relative scatter of 1.018 looks convicing. However, there are significant variations of this quantity. At wavelengths where the continuum error is high, the scatter is too high as well. On the other hand, the variation of the OH bands modelled by a single climatology (without significant solar cycle effect) appears to be underestimated.

This issue seems to extend to the NIR-arm range (c), where even the average scatter decreases to 0.762 for the range up to 1.78 μm. Apart from the OH bands, the $O_2(a\text{-}X)(0\text{-}0)$ band near 1.27 μm is affected. Note that Noll et al. (2012) did not have the data to measure the variability of this band and therefore used the results for $O_2(b\text{-}X)(0\text{-}1)$, which is quite different as (d) and (e) in Fig. 8 illustrate. The $HO_2$ variability in (g) is also very different from the situation near 543 nm. The mean relative deviation also indicates an underestimation between 1.02 and 1.78 μm ($-0.182$). As the structures in Fig. 11c are relatively similar to those of the relative scatter, the same emissions are affected. The deviations are probably related to the extrapolation of the reference intensities derived from the FORS 1 data, which relied on the simple population model of Rousselot et al. (2000) for OH (see Sect. 1). As this model only assumes cold rotational populations, the hot population lines are far too weak. Moreover, the use of old $A$ coefficients from HITRAN2008 (Rothman et al., 2009) could have had an impact. As the modelled $O_2(a\text{-}X)$ emissions were scaled to the OH emissions based on the small sample of X-shooter spectra (Noll et al., 2014), significant deviations are also present, at least for the strong 1.27 μm band. At the position of the weak 1.58 μm band, there is no clear offset.

## 8   Conclusions

The pioneering ESO Sky Model for Cerro Paranal in Chile has been very important with respect to astronomical observing proposal evaluation, observation scheduling, data processing, and the design of new instruments. As discussed in the previous section, the crucial airglow component is, however, affected by systematic issues due to the lack of suitable data for the development of the model by Noll et al. (2012). The data situation has improved a lot in the past decade, especially by the availability of a well-calibrated X-shooter data set of the order of $10^5$ spectra covering the wide wavelength from 0.3 to 1.8 μm and a period of 10 years. Moreover, the authors of this study carried out several studies that significantly improved the understanding of airglow line and pseudo-continuum emissions at Cerro Paranal. Therefore, the preconditions to build





a significantly better model are fulfilled. As a consequence, we created the Paranal Airglow Line And Continuum Emission (PALACE) model. It consists of a comprehensive line list with 26,541 entries, 3 continuum components, and two-dimensional climatologies of relative intensity, solar cycle effect, and residual variability for 23 variability classes. The model is provided in connection with a Python/Cython code for the calculation of airglow spectra (with uncertainty estimates) depending on
different input parameters. The comparison of the model and the observed X-shooter spectra (where the other sky radiance components were still removed by means of the ESO Sky Model) showed a convincing agreement with average uncertainties in terms of the mean difference and the scatter of only a few per cent. This performance should be appealing for the use of PALACE in similar contexts as the current ESO Sky Model.

The development of PALACE also resulted in new insights in airglow physics and chemistry. For the OH line list, a com-
prehensive population model with one to three temperature components for the vibrational levels $v$ from 2 to 10 was created based on fits of X-shooter data as well as UVES data from previous studies (Cosby et al., 2006; Noll et al., 2020). In this context, also the input Einstein-$A$ coefficients were further improved compared to Noll et al. (2020). Population models were also created for the excited $a^1\Delta_g(v \leq 1)$ and $b^1\Sigma_g^+(v \leq 2)$ states of $O_2$ based on new X-shooter measurements near 0.865 and 1.27 μm and UVES data from Cosby et al. (2006). A model for intensities of O recombination lines (e.g. at 777 nm) was
derived from the combination of X-shooter-based intensities, theoretical recombination coefficients, and $A$ coefficients. The measurements of atomic lines also include green O, red O, Na D, N 520 nm, and several H Balmer lines. From Noll et al. (2019), the UVES-based time series of the K 769.9 nm line was taken. For the derivation of the continuum model consisting of the components of $HO_2$, FeO (plus other molecules), and unresolved $O_2$ at short wavelengths, we used the X-shooter-based results of Noll et al. (2024a).

Climatologies from the time series data of 392 different emissions were considered for the PALACE variability model consisting of 23 classes. Extending the studies of Noll et al. (2019, 2023b, 2024a), the relative intensity climatologies of emissions in the mesopause region can be grouped depending on the most crucial chemical species for the variations, i.e. either H, O, or $O_3$. The intensities of ionospheric lines mainly depend on the electron/ion density, whereas H fluorescence emissions are strongly influenced by the height of the Earth's shadow. The comparison of two-dimensional climatologies of month and
local time for the solar cycle effect and residual variability (including the effective nighttime values) also revealed interesting similarities and differences which were not investigated before. The results show the height-dependent impact of wave-like perturbations such as tides (especially the migrating diurnal tide), gravity waves (especially long periods), and planetary waves (especially the quasi-two-day wave) on the climatological pattern. For ionospheric lines, the change of the vertical ionisation distribution is important, where waves and instabilities can lead to additional variability. As the main goal of this study was the
creation of PALACE, a more detailed study of the time series and climatological data was out of the focus. Hence, the large data set should be valuable for further investigations.

*Code and data availability.* The Python/Cython code of PALACE v1.0, the necessary model files in FITS table format, the reference test output, and different supplementary ASCII tables for a better understanding of the model and the reproduction of the figures are provided



at https://zenodo.org/records/14064023 (Noll et al., 2024b) under a CC-BY-4.0 licence for the data as well as a GNU GPLv3 licence for the code. In the case of future versions of PALACE, the paper-related release files will remain untouched, but a link to the revised model will be provided in the metadata. Apart from own measurements, the study also used X-shooter-based data of previous publications (Noll et al., 2023b, 2024a) released via Zenodo (Noll et al., 2023c, a). The basic X-shooter data for this project originate from the ESO Science Archive Facility at http://archive.eso.org and are related to various observing programmes that were carried out between October 2009 and September 2019. Moreover, UVES-related data sets jointly published with the articles by Cosby et al. (2006), Noll et al. (2019), and Noll et al. (2020)

were considered for the model development. The raw UVES data and more processed Phase 3 spectra are also provided by the ESO Science Archive Facility.

*Author contributions.* SN designed and organised the project, performed the the analysis of the data, derived and tested the model, wrote the code, created figures and tables, and is the main author of the paper text. The co-authors (especially CS) contributed to the paper content. Moreover, PH and WK tested the code. WK performed the basic processing of the X-shooter spectra. SK managed the infrastructure for the
processing and storage of the X-shooter data.

*Competing interests.* Competing interests are not present.

*Acknowledgements.* The work of Stefan Noll was partly financed by the project NO 1328/1-3 of the German Research Foundation (DFG). We are grateful to Michael Bittner from DLR for his support in terms of the infrastructure for the project and the funding of this publication. Moreover, we thank Sabine Möhler from ESO for her support with respect to the X-shooter calibration data.



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
