# Peer review of "PALACE v1.0: Paranal Airglow Line And Continuum Emission model"

_EGUsphere, 2024_

## Author Response (AR1)

**Response to 'Comment on egusphere-2024-3512' by Anonymous Referee #1**

"It is an excellent review of the entire topic."

"The writing is exceptional for clarity."

"This GMD paper is a tour de force. It has improved our understanding of the upper atmosphere. I see no problems with publishing it as is."

**We are impressed by the touching words of Anonymous Referee #1. This review makes us confident that several years of hard work have really led to something very valuable for airglow research and optical astronomy.**

**Response to 'Comment on egusphere-2024-3512' by Anonymous Referee #2**

The manuscript includes many details on how the new model is developed and physical insights during the development. I think the manuscript is well-written.

**We are pleased that Anonymous Referee #2 appreciates our thorough work on the model and the manuscript.**

I have several suggestions to improve its readability.

**We thank the reviewer for the helpful proposals for an improvement of the comprehensibility of the model and its discussion.**

1. As the authors described, the development of PALACE benefits from the X-shooter data. I think it is better to give an introduction to X-shooter and its data, and make it clearer that how the X-shooter is useful for the PALACE development.

**In Sect. 3, we introduce the X-shooter echelle spectrograph and the related data set in detail. As it appears that this information can be overlooked in the long manuscript, we have especially changed the summary of the contents at the end of Sect. 1. Instead of *"Section 3 provides details on the data set for the analysis"*, it now reads *"Section 3 provides details on the X-shooter and UVES spectrographs and the related data sets that were used for the development of PALACE"*. Furthermore, we have changed some expressions in Sect. 3 to underline the uniqueness of the X-shooter data set.**

2. A flowchart should be given to show how different parts of the model are connected, and how the user inputs are converted to output spectra. This may help readers quickly grasp the overall structure of the model.

**Yes, this is a good idea in order to complement the discussion in Sect. 2 by a visualisation of the structure of the code. This will certainly help the readers to easier understand how the model is calculated. Our proposed flowchart is attached to this response. It will be added to the paper as a new Fig. 1 with the caption**

*"Flowchart of PALACE executed as command-line script. By loading the Python module 'palace', the displayed functions can be called directly. The details of the code structure are discussed in the text."*

**Moreover, there will be several references to this figure in Sect. 2.**

[Figure]

3. The manuscript does not show how the PALACE model can be used to improve astronomical observations. It would be better to give one or two examples and show the new PALACE can reduce biases in observations compared with the traditional Sky Model.

**Indeed, we kept the discussion of possible applications of the model very short as we focused on the description of the structure and development of the model, which already led to a long manuscript. However, we recognise that astronomers will probably constitute a large fraction of the users of the code. Hence, some explanations with respect to the potential impact of**

**PALACE on astronomical applications are really useful. Therefore, we have added three new paragraphs at the end of Sect. 7:**

*"Sky-brightness models are crucial for an efficient scheduling of the telescope operations at large astronomical observatories (Noll et al., 2012). Time estimates are obtained via exposure time calculators (ETCs). For the ESO sites in Chile, see https://etc.eso.org/. In the case of overestimations, precious observing time can be wasted, whereas the opposite case can lead to data of too low quality for the scientific goals. As the time predictions are usually made in advance (especially during the application process for the observing time), there are unavoidable uncertainties for individual observations as the actual atmospheric conditions are not considered. However, the overall performance of an ETC will clearly depend on the quality of the sky-brightness model. Hence, the discussed shortcomings of the current ESO Sky Model with respect to airglow emission can have a significant impact on the efficiency of the telescope operations.*

*In the near-IR range shown in Fig. 12c, airglow is clearly dominating the night-sky brightness. Even scattered light from the Moon is only a minor contribution in this wavelength regime (Jones et al., 2019). Hence, the underestimations of the airglow radiance and its variability by the ESO Sky Model should significantly affect the performance of the ETC. In the wavelength range from 1.02 to 1.78 μm, 23% of the bins show deviations that are larger than 50% of the X-shooter-related flux. Assuming an astronomical object that is much fainter than the airglow emission, the signal-to-noise ratio is approximately proportional to the ratio of the detector electron counts from the object and the square root of those from the sky ($N\_obj / sqrt(N\_sky)$) if read-out noise and dark currents can be neglected. Hence, a relative deviation of -0.5 in Fig. 12c would correspond to an overestimation of the signal-to-noise ratio by a factor of about 1.4. In order to achieve the same real quality of the data, the exposure time would need to be doubled, which would have a major impact on the scheduling of the observations. The opposite case (i.e. an overestimation of the required exposure time) is expected for the airglow continuum at wavelengths of about 920 nm, where Fig. 12b indicates a flux of the ESO Sky Model that is about 3 times higher than expected. As the airglow continuum only provides about 50% of the total sky brightness at these wavelengths for the analysed X-shooter data on average, the exposure time could be decreased by a factor of about 1.5 for our simplified scenario in dark nights without a bright Moon.*

*In conclusion, the distinctly better performance of PALACE with respect to the prediction of the airglow brightness should also have a significant impact on the quality of astronomical exposure time calculations and the planning of telescope operations. Moreover, we expect an influence on the development of new astronomical instruments as the sky brightness is also a basis for the design of instruments in the context of the corresponding scientific goals. The brightness and variability of airglow also matters for the processing of astronomical data. In particular, the ESO Sky Model is used for the removal of airglow lines in one-dimensional spectra (Noll et al., 2014). Although the subtraction is based on an observed spectrum scaled by wavelength-dependent factors, the model is required for the identification of lines and the initial weights of different variability groups to each pixel. Here, the PALACE model data should also allow for improvements."*

**Moreover, we have added a sentence at the end of the first paragraph in Sect. 8:**

*"With respect to astronomical observatories, such applications are the evaluation of observing proposals, the scheduling of telescope time, the design of instruments, and algorithms for data processing."*

4. Line 40, 'hight' may be 'height'.

**We have corrected the typo.**